# Genetic associations of protein-coding variants in human disease

Benjamin B. Sun[1,2✉], Mitja I. Kurki[3,4,5,6], Christopher N. Foley[7,8], Asma Mechakra[9], Chia-Yen Chen[1], Eric Marshall[1], Jemma B. Wilk[1], Biogen Biobank Team*, Mohamed Chahine[10], Philippe Chevalier[9], Georges Christé[9], FinnGen*, Aarno Palotie[3,4,5,6], Mark J. Daly[3,4,5,6] & Heiko Runz[1✉]

Genome-wide association studies (GWAS) have identified thousands of genetic variants linked to the risk of human disease. However, GWAS have so far remained largely underpowered in relation to identifying associations in the rare and low-frequency allelic spectrum and have lacked the resolution to trace causal mechanisms to underlying genes[1]. Here we combined whole-exome sequencing in 392,814 UK Biobank participants with imputed genotypes from 260,405 FinnGen participants (653,219 total individuals) to conduct association meta-analyses for 744 disease endpoints across the protein-coding allelic frequency spectrum, bridging the gap between common and rare variant studies. We identified 975 associations, with more than one-third being previously unreported. We demonstrate population-level relevance for mutations previously ascribed to causing single-gene disorders, map GWAS associations to likely causal genes, explain disease mechanisms, and systematically relate disease associations to levels of 117 biomarkers and clinical-stage drug targets. Combining sequencing and genotyping in two population biobanks enabled us to benefit from increased power to detect and explain disease associations, validate findings through replication and propose medical actionability for rare genetic variants. Our study provides a compendium of protein-coding variant associations for future insights into disease biology and drug discovery.

Inherited variations in protein-coding and non-coding DNA have a role in the risk, onset and progression of human disease. Traditionally, geneticists have dichotomized diseases as either caused by coding mutations in single genes that tend to be rare, highly penetrant and frequently compromise survival and reproduction (often termed 'Mendelian' diseases), or as common diseases that show a complex pattern of inheritance influenced by the joint contributions of hundreds of low-impact, typically non-coding genetic variants (often termed 'complex' diseases). For both rare and common conditions, large human cohorts systematically characterized for a respective trait of interest have enabled the identification of thousands of disease-relevant variants through either sequencing-based approaches or GWAS. Nevertheless, the exact causal alleles and mechanisms that underlie associations of genetic variants to disease have so far remained largely elusive[1].

In recent years, population biobanks have been added to the toolkit for disease gene discovery. Biobanks provide the opportunity to simultaneously investigate multiple traits and diseases at once and uncover relationships between previously unconnected phenotypes. For instance, the UK Biobank (UKB) is a resource that captures detailed phenotype information matched to genetic data for more than 500,000

individuals and, since its inception, has facilitated biomedical discoveries at an unprecedented scale[2]. We and others have recently reported on the ongoing efforts to sequence the exomes of all UKB participants and link genetic findings to a broad range of phenotypes[3–6]. We also established FinnGen (FG) (https://www.finngen.fi), an academic–industry collaboration to identify genotype–phenotype correlations in the Finnish founder population with the aim of better understanding how the genome affects health. Finland is a well-established genetic isolate and a unique gene pool distinguishes Finns from other Europeans[7]. The distinct Finnish haplotype structure is characterized by large blocks of co-inherited DNA in linkage disequilibrium and an enrichment for alleles that are rare in other populations, but can still be confidently imputed from genotyping data even in the rare and ultra-rare allele frequency spectrum[8–10]. Through combining imputed genotypes with detailed phenotypes ascertained through national registries, FG holds the promise to provide particular insights into the so far little examined allele frequency spectrum between 0.1 and 2%, where both sequencing studies and GWAS have so far remained largely underpowered in relation to identifying associations with disease. This spectrum includes many coding variants with moderate to large effect sizes that can help

[1]Translational Biology, Research and Development, Biogen Inc., Cambridge, MA, USA. [2]BHF Cardiovascular Epidemiology Unit, Department of Public Health and Primary Care, University of Cambridge, Cambridge, UK. [3]Psychiatric and Neurodevelopmental Genetics Unit, Massachusetts General Hospital, Boston, MA, USA. [4]The Stanley Center for Psychiatric Research, The Broad Institute of MIT and Harvard, Cambridge, MA, USA. [5]Institute for Molecular Medicine Finland (FIMM), University of Helsinki, Helsinki, Finland. [6]Analytic and Translational Genetics Unit, Department of Medicine, Massachusetts General Hospital, Boston, MA, USA. [7]MRC Biostatistics Unit, School of Clinical Medicine, University of Cambridge, Cambridge, UK. [8]Optima Partners, Edinburgh, UK. [9]Université de Lyon 1, Université Lyon 1, INSERM, CNRS, INMG, Lyon, France. [10]CERVO Brain Research Center and Department of Medicine, Faculty of Medicine, Université Laval, Quebec City, Quebec, Canada. *Lists of authors and their affiliations appear at the end of the paper. ✉e-mail: bbsun92@outlook.com; heiko.runz@gmail.com

identify causal genes in GWAS loci, provide mechanistic insights into disease pathologies, and potentially bridge rare and common diseases.

Here, we have leveraged the combined power of UKB and FG to investigate how rare and low-frequency variants in protein-coding regions of the genome contribute to the risk for human traits and diseases. Using data from a total of 653,219 individuals, we tested how approximately 48,000 coding variants identified in both biobanks through either whole-exome sequencing or genotype imputation associate with 744 distinct disease endpoints. Disease associations were compared against information from rare disease, biomarker and drug target resources and complemented by deep dives into distinct disease mechanisms of individual genes and coding variants. Our results showcase the benefits of combining large population cohorts to discover and replicate novel associations, explain disease mechanisms across a range of common and rare diseases, and shed light on a substantial gap in the allelic spectrum that neither genotyping nor sequencing studies have previously been able to address.

## Coding associations with human disease

An overview of the study design and basic demographics are provided in Extended Data Fig. 1, Supplementary Table 1. In brief, we systematically harmonised disease phenotypes across UKB and FG using Phecode and ICD10 mappings and retained 744 specific disease endpoints grouped into 580 disease clusters (Methods, Supplementary Table 2). Disease case counts relative to cohort size showed good correlations both overall between UKB and FG (Spearman's $\rho = 0.65$, $P < 5.3 \times 10^{-90}$) and across distinct disease groups (Extended Data Fig. 2).

We performed coding-wide association studies (CWAS) across 744 disease endpoints over a mean of 48,189 (range: 25,309–89,993) (Methods, Supplementary Table 2) post-quality control coding variants across the allele frequency spectrum derived from whole-exome sequencing of 392,814 European ancestry individuals in UKB and meta-analysed these data with summary results from up to 260,405 individuals in FG (Methods, Supplementary Table 2).

We identified 975 associations (534 variants in 301 distinct regions across 148 disease clusters; 620 distinct region–disease cluster associations) meeting genome-wide significance ($P < 5 \times 10^{-8}$), and 717 associations (378 variants in 231 distinct regions across 121 disease clusters; 445 distinct region–disease cluster associations) at a conservative (Bonferroni) multiple testing threshold of $P < 2 \times 10^{-9}$ (correcting for the number of approximate independent tests) (Methods, Fig. 1a, Supplementary File 1 (interactive), Supplementary Table 3). The distributions of coding variant annotation categories were largely similar for variants with at least one significant association ($P < 5 \times 10^{-8}$) relative to all variants tested, with missense variants showing a higher fraction of significant variants than in-frame insertion–deletions or predicted loss-of-function (pLOF) variants (Extended Data Fig. 3). Inflation was well controlled with a mean genomic inflation factor of 1.04 (5th–95th percentiles: 1.00–1.09; Extended Data Fig. 4a). Effect sizes were generally well aligned between UKB and FG (Spearman's $\rho = 0.90$, $P < 10^{-300}$) (Extended Data Fig. 4b). Minor allele frequencies (MAFs) of lead variants correlated well overall between UKB and FG (Spearman's $\rho = 0.97$, $P < 10^{-300}$) (Fig. 1b), especially for variants with MAF > 1%, yet as expected[9] from genetic differences between Finns and non-Finnish Europeans (NFEs) this correlation was reduced for variants with MAF < 1% (Spearman's $\rho = 0.32$, $P = 0.023$).

Across all diseases, we found generally larger effect sizes for low frequency and rare variants (Fig. 1c). Of the 975 identified associations, 387 (39.7% at $P < 5 \times 10^{-8}$, 270 out of 717 (37.7%) at $P < 2 \times 10^{-9}$) would not have been detected if analysed in UKB (61.5% at $P > 5 \times 10^{-8}$; 60.1% at $P > 2 \times 10^{-9}$) or FG (59.6% at $P > 5 \times 10^{-8}$; 58.6% at $P > 2 \times 10^{-9}$) alone. We found 13 associations (across 11 genes) with log odds ratios greater than 2 (Fig. 1c). Of these, 12 associated variants had MAF < 1%, and only the haemochromatosis variant rs1800562 showed frequency

ranges traditionally interrogated in GWAS (MAF of 7.9% (UKB) and 3.7% (FG)). Several variants with large effect sizes reside in well studied disease genes such as *BRCA1* (breast cancer), *IDH2* (myeloid leukaemia), *VWF* (von Willebrand disease) or *HFE* (disorders or iron metabolism), proposing that carriers could benefit from clinical monitoring for associated conditions. Association testing within UKB and FG individually would have yielded 318 and 479 associations, respectively, at $P < 5 \times 10^{-8}$ (Supplementary Tables 4, 5). Thus, our combined approach using both biobanks increased the number of significant findings by approximately threefold for UKB and twofold for FG. Of the 318 and 479 significant sentinel variants in UKB and FG, 252 (72.6%) and 258 (53.9%) replicated at $P < 0.05$ in FG and UKB, respectively (Supplementary Tables 4, 5), further highlighting the strength of our approach to yield results that are more robust through replication than findings derived from a single biobank.

Our study benefits from population enrichment of rare alleles in Finns versus NFEs (and vice versa) that increases the power for association discovery. Using a combination of theoretical analyses and empirical simulations, we show that by leveraging population-enriched variants we could increase inverse-variance weighted meta-analysis $Z$-scores and hence our ability to detect underlying associations (Supplementary Information). The gain in power from enriched alleles was present across a range of rare MAFs (0.01–1%), with the strongest power gain in the rare and ultra-rare MAF range of 0.01% to 0.25% (Fig. 1d, Extended Data Fig. 5, Supplementary Information, Supplementary Files 2a–c (interactive)). Notably, we demonstrate both theoretically and in practice that gains in power due to allele enrichment remain even after adjusting for power gains due to increased sample size (Supplementary Fig. 2 (MAF enrichment on $Z$-scores), Extended Data Fig. 5d). Of the sentinel variants, we found 73 (33 in UKB and 40 in FG) to be enriched by more than twofold and 23 (8 in UKB and 15 in FG) by more than fourfold relative to the other biobank (Fig. 1b, Supplementary Table 6). Most highly population-enriched variants are rare (MAF<1%) or low frequency (MAF 1–5%), whereby 20 out of 23 variants with more than fourfold population enrichment (13 in FG and 7 in UKB) had MAF <1% (Table 1, Supplementary Table 6). In comparison, 52 out of the total of 534 (9.7%) sentinel variants had MAF<1% in either UKB or FG, of which 15 and 23 were enriched by more than twofold in UKB and FG, respectively (Supplementary Table 6).

We systematically cross-referenced our results with previously described GWAS associations (via GWAS Catalog[11] and PhenoScanner[12]) and disease relevance as reported in ClinVar[13] (Methods). In total, we found that 216 out of 620 (34.8%) distinct region–disease cluster associations (at $P < 5 \times 10^{-8}$) had not previously been reported (130 out of 445 (29.2%) at $P < 2 \times 10^{-9}$). Out of the 216 distinct loci, 177 (104 out of 130 at $P < 2 \times 10^{-9}$) were in genes not previously mapped to the respective diseases (Fig. 1a, Supplementary Table 3, Supplementary File 1 (interactive)). Of the novel associations at GWAS significance ($P < 5 \times 10^{-8}$), roughly one-third had MAF < 5% in either UKB or FG and 15% had MAF < 1% (Supplementary Table 3). Notably, in UKB, 17% of known (19% in FG) and 31% of novel (28% in FG) associations had a MAF < 5%. Correspondingly, in UKB, 5% of known (6% in FG) and 15% of novel (10% in FG) associations had a MAF < 1%, highlighting the power gained through our approach especially in the low and rare allele frequency spectrum (Fig. 1e, Supplementary Table 3).

Mapping associations to genes, we found a total of 482 unique genes associated with the 148 disease clusters. Approximately 92% of the associated regions for each disease cluster (excluding the major histocompatibility complex (*MHC*) cluster) harbour a single gene with coding associations (Extended Data Fig. 6a). The majority of gene loci (81.2% at $P < 5 \times 10^{-8}$; *MHC* region counted as one locus) were associated with a single disease cluster (Extended Data Fig. 6b). Thirteen loci were associated with at least five trait clusters (at $P < 5 \times 10^{-8}$), including well established pleiotropic regions such as the *MHC*, *APOE*, *PTPN22*, *GCKR*, *SH2B3* and *FUT2* (Fig. 1a). For instance, in addition to a known

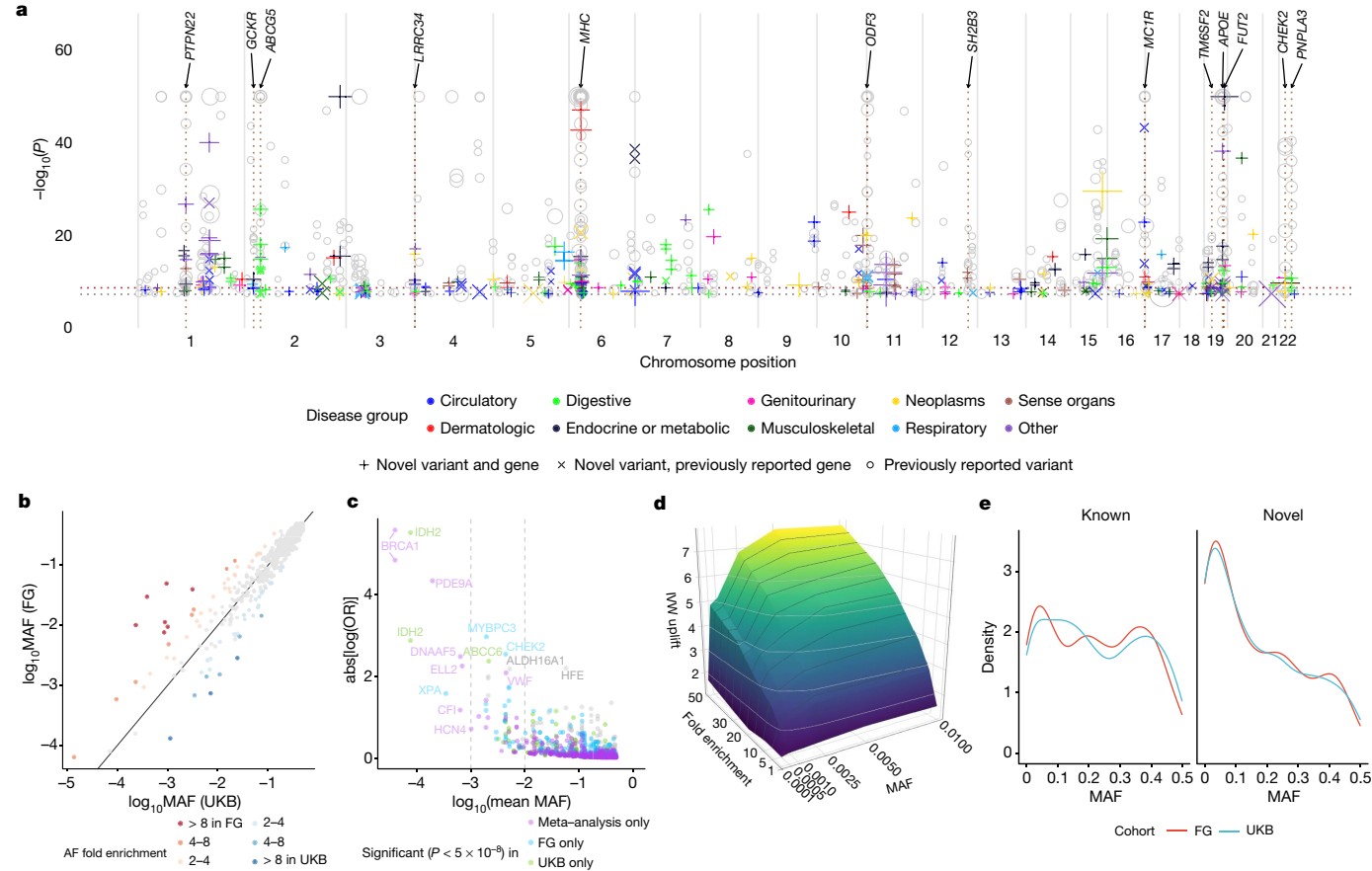

**Fig. 1 | Coding genetic associations with disease. a**, Summary of sentinel variant associations. Size of the point is proportional to effect size. $-\log_{10}(P)$ capped at $-\log_{10}(10^{-50})$. Labels highlight pleiotropic associations ($\geq$5 trait clusters). Colours indicate disease groups. Shapes indicate novel and known (grey circles) associations. Dotted horizontal lines indicate $-\log_{10}(2\times10^{-9})$ (brown) and $-\log_{10}(5\times10^{-8})$ (grey). **b**, Comparison of sentinel variant MAF between UKB and FG data. **c**, Effect size against MAF of sentinel variants. Dashed lines indicate MAF of 0.1% (left) and 1% (right). Genes for coding variant

associations with absolute effect size greater than 2 or MAF less than 0.1% are labelled. **d**, Surface plot of effects of cohort specific allele enrichment on inverse variant weighted (IVW) meta-analysis $z$-scores (IVW uplift) across MAFs (up to MAF 1%). Uplift is defined as the ratio of meta-analysed IVW $z$-score to the $z$-score of an individual study (details in Supplementary Information). **e**, Density plot of MAF for sentinel variants for known versus novel associations. Interactive Manhattan plot for novel associations and allelic enrichment surface plots are provided as Supplementary Files 1, 2.

association with breast cancer, we found variants in *CHEK2* to be associated with the risk of colorectal and thyroid cancers, uterine leiomyoma, benign meningeal tumours and ovarian cysts. Also, in addition to a known association with prostate hyperplasia, we found an *ODF3* missense variant (rs72878024, MAF = 7.5% (UKB) and 7.7% (FG)) to be associated with risk of uterine leiomyoma, benign meningeal tumour, lipoma and polyps in the female genital tract (Supplementary Table 3).

Harnessing the added power of UKB and FG, we were able to detect GWAS associations for rare variants previously only annotated as causal for single-gene diseases, establishing a disease relevance for these variants at the population level. Of the 534 distinct variants with significant disease associations in our study ($P < 5\times10^{-8}$), 152 (28.5%) had previously been linked to diseases in ClinVar. For 46 (30.3%) of these variants, the associated disease cluster matched with a previously reported phenotype in ClinVar. Notably, only 7 of these 46 variants (in *GJB2*, *ABCC6*, *BRCA1*, *SERPINA1*, *FLG*, *IDH2* and *MYOC*) had a previous annotation as either pathogenic or probably pathogenic (Supplementary Table 7), with 15 others annotated as benign. For the remaining 106 ClinVar-listed variants, 29 (27.4%) showed associations to conditions putatively related to those listed in ClinVar (Supplementary Table 7, Methods). For 17 variants, the medical relevance had been reported in ClinVar for the associated conditions, with 3 being classified as pathogenic or probably pathogenic and 14

classified either as benign or having 'conflicting interpretation of pathogenicity' for the associated trait (Supplementary Tables 3, 7). For instance, we found a rare missense variant annotated as showing conflicting pathogenicity in ClinVar in *VWF* (rs1800386:C; Tyr1584Cys; MAF = 0.44% (UKB) and 0.47% (FG)) to be associated with the risk of von Willebrand disease[13] (log(odds ratio (OR)) = 2.09, $P = 8.7\times10^{-9}$). We also assessed the medical actionability of associated genes as defined in the latest American College of Medical Genetics and Genomics (ACMG) guidelines[14] and found 15 coding variants with significant associations in 11 ACMG genes (Supplementary Table 7). Thirteen of these associations (one pathogenic (*BRCA1*), four conflicting evidence of pathogenicity and eight benign or probably benign) had prior ClinVar reports to a matching or putatively related condition, and for several our results proposed extended phenotypes. For example, we found that carriers of the rare missense variant rs370890951 (Ile1131Thr; MAF = 0.097% (UKB) and 0.29% (FG)) in *MYBPC3*, in which mutations cause hypertrophic cardiomyopathy, showed an approximately threefold increased risk ($P = 9.8\times10^{-13}$) for coagulation defects (Supplementary Tables 3, 7). Together, these findings highlight that population-scale analyses like ours can help refine pathogenicity assignments through contributing quantitative, rather than qualitative, information on relative disease risks for variant carriers, or establish an 'allelic series' for medically actionable genes.

**Table 1 | Genes with sentinel variants enriched more than fourfold in either UKB or FG datasets**

| Gene | rs ID (amino acid change) | Chr | $A_0/A_1$ | $A_1$ frequency (UKB; FG (%)) | $\log_2$ FE (FG/UKB) | OMIM gene–phenotype relationships | CWAS gene–phenotype relationships |
|---|---|---|---|---|---|---|---|
| CHEK2 | rs17879961 (I200T)<br>rs555607708[a] (T410fs) | 22 | A/G<br>AG/– | 0.04%; 2.99%<br>0.24%; 0.64% | 6.25<br>1.42 | Cancer (breast, prostate, colorectal, osteosarcoma); Li–Fraumeni syndrome | rs17879961:G, benign meningeal neoplasm<br>rs555607708:del (2.8× FG enriched), cancer (breast, thyroid, colorectal (benign)); uterine leiomyoma; ovarian cysts; PCOS |
| DBH | rs77273740 (R79W) | 9 | C/T | 0.10%; 4.95% | 5.69 | Orthostatic hypotension | Hypertension (IA) |
| PITX2 | rs143452464 (P41S) | 4 | G/A | 0.02%; 1.01% | 5.42 | Anterior segment dysgenesis; Axenfeld–Rieger syndrome; ring dermoid of cornea | Arrythmia and AF |
| SLC24A5 | rs1426654 (T111A) | 15 | A/G | 0.09%; 1.13% | 3.67 | Skin, hair, eye pigmentation (dark); oculocutaneous albinism | Non-epithelial cancer of skin (other) (IA) |
| CFHR5 | rs565457964 (E163fs) | 1 | C/CAA | 0.32%; 3.96% | 3.66 | Nephropathy due to CFHR5 deficiency | Degeneration of macula and posterior pole of retina (IA) |
| ANKH | rs146886108 (R187Q) | 5 | C/T | 0.72%; 0.07% | -3.28 | Chondrocalcinosis; craniometaphyseal dysplasia | Type 2 diabetes mellitus (IA) |
| ALDH16A1 | rs150414818 (P527R) | 19 | C/G | 0.10%; 0.95% | 3.23 | – | Gout |
| LRRK1 | rs41531245 (T967M) | 15 | C/T | 0.09%; 0.76% | 3.15 | – | Contracture of palmar fascia; fasciitis; umbilical hernia |
| CFI | rs141853578 (G119R) | 4 | C/T | 0.11%; 0.01% | −3.10 | Atypical haemolytic uremic syndrome; age-related macular degeneration; CFI deficiency | Retinal disorders (other) |
| FLG | rs61816761 (R501*)<br>rs138381300[a] (S761fs) | 1 | G/A<br>CACTG/– | 2.45%; 0.29%<br>2.45%; 1.35% | −3.10<br>−0.85 | Atopic dermatitis; ichthyosis vulgaris | rs61816761:A, dermatitis (other)<br>rs138381300:del (1.8× UKB enriched), asthma; non-epithelial cancer of skin (other) |
| SOS2 | rs72681869 (P191R) | 14 | G/C | 1.09%; 0.15% | -2.84 | Noonan syndrome | Hypertension (IA) |
| XPA | rs144725456 (H244R) | 9 | T/C | 0.01%; 0.06% | 2.61 | Xeroderma pigmentosum | Non-epithelial cancer of skin (other) |
| CDC25A | rs146179438 (Q24H) | 3 | C/A | 1.52%; 8.72% | 2.52 | – | Kidney and urinary stones (IA) |
| F10 | rs61753266 (E142K) | 13 | G/A | 0.33%; 1.83% | 2.46 | Factor X deficiency | PE and pulmonary heart disease (inverse association) |
| TNXB | rs61745355 (G2848R)<br>rs10947230[a] (R2704H)<br>rs11507521 (T302A) | 6 | C/T<br>C/T<br>T/C | 2.22%; 11.86%<br>5.96%; 14.75%<br>13.29%; 9.17% | 2.42<br>2.31<br>−0.54 | Ehlers–Danlos syndrome; vesicoureteral reflux | rs61745355:T, lymphoma<br>rs10947230:T, lichen planus<br>rs1150752:C, chronic hepatitis; other inflammatory liver diseases; atherosclerosis |
| SLC39A8 | rs13107325 (A391T) | 4 | C/T | 7.40%; 1.46% | −2.35 | Congenital disorder of glycosylation | Shoulder lesions |
| CLPTM1 | rs150484293 (L140F) | 19 | C/T | 0.35%; 0.07% | −2.33 | – | Dementia |
| ELL2 | rs141299831 (S18L) | 5 | G/A | 0.02%; 0.12% | 2.29 | – | Benign neoplasm of other and ill-defined parts of digestive system |
| CASP7 | rs141266925 (F214L) | 10 | T/C | 0.31%; 1.5% | 2.29 | – | Cataracts |
| BRCA1 | rs80357906 (Q1777fs) | 17 | T/TG | 0.001%; 0.01% | 2.21 | Cancer (breast, ovarian, pancreatic); Fanconi anaemia | Breast cancer |
| SCN5A | rs45620037 (T220I) | 3 | G/A | 0.11%; 0.49% | 2.20 | Sudden infant death syndrome; dilated cardiomyopathy; arrythmia[b] | Arrythmia and AF |
| CACNA1D | rs1250342280 (F1943del) | 3 | CCTT/C | 0.60%; 0.14% | −2.09 | Primary aldosteronism, seizures, and neurologic abnormalities; sinoatrial node dysfunction and deafness | Hypertension |
| WNT10A | rs121908120 (F228I) | 2 | T/A | 2.72%; 0.65% | −2.06 | Odontoonychodermal dysplasia; Schopf–Schulz–Passarge syndrome; selective tooth agenesis | Follicular cysts of skin and subcutaneous tissue (IA) |

[a]Other sentinel variants in the gene with greater than fourfold enrichment.

[b]Sudden infant death syndrome; atrial fibrillation; Brugada syndrome; progressive and non-progressive heart block; long QT syndrome, sick sinus syndrome; ventricular fibrillation.

All enrichment indicated by two-sided Fisher's test; unadjusted $P < 5 \times 10^{-5}$.

AF, atrial fibrillation; Chr, chromosome; FE, fold enrichment; IA, inverse association; PCOS, polycystic ovarian syndrome; PE, pulmonary embolism; $A_0$, reference allele; $A_1$, alternative (effect) allele.

Seventeen of the twenty-three genes with highly population-enriched sentinel variants (Table 1) were listed as disease genes at Online Mendelian Inheritance in Man (OMIM). Of these, ten (*CHEK2*, *DBH*, *SCL24A5*, *CFI*, *FLG*, *XPA*, *F10*, *BRCA1*, *SCN5A* and *CACNA1D*) showed associations with conditions identical or related to the respective Mendelian disease, revealing a relevance of the associated variants on the population level. For instance, we found the missense variant rs77273740 in *DBH* (enriched by more than 50-fold in FG)—a gene associated with orthostatic hypotension—to be associated with reduced risk of hypertension (log(OR) = −0.19, $P = 1.3 \times 10^{-23}$), and an in-frame deletion (rs1250342280) in *CACNA1D* (enriched by 4.3× in UKB)—a gene associated with primary aldosteronism—was associated with increased risk of hypertension (log(OR) = 0.19, $P = 2.0 \times 10^{-8}$) (Table 1).

## Biomedical insights through CWAS

We leveraged the coding variant associations identified in our study to generate biological insights for a range of distinct genes, pathways and diseases and in the following exemplify the broad utility of our resource with a set of selected use cases.

### Coagulation proteins in pulmonary embolism

We found known and novel associations with pulmonary embolism risk, including two rare variant associations (average MAF < 1%) in genes encoding components of the coagulation cascade at the convergent common pathway (Extended Data Fig. 7). For instance, we discovered a rare missense mutation in *F10*, enriched by approximately fivefold in FG (rs61753266:A; Glu142Lys; MAF = 0.33% (UKB) and 1.85% (FG)), and a venous thromboembolism risk-reducing variant in *F5* (rs4525:C, His865Arg; MAF = 27.2% (UKB) and 22.3% (FG)), to be protective against pulmonary embolism (log(OR)$_{F10}$ = −0.44, $P_{F10} = 2.9 \times 10^{-9}$; log(OR)$_{F5}$ = −0.14, $P_{F5} = 1.2 \times 10^{-15}$). The effects of these associations on the levels of their respective circulating factors and thromboembolic diseases, Mendelian randomization analyses that support developing drugs inhibiting factors V and X for pulmonary embolism and findings on additional clotting factors are discussed in (Supplementary Information 'New roles for coagulation proteins in PE').

### Rare variants yield mechanistic insights

We interrogated the sentinel variants identified in this study for associations with 117 quantitative biomarkers spanning eight categories in UKB (Supplementary Table 8). At a multiple testing adjusted threshold of $P < 1 \times 10^{-6}$, we found 112 of the biomarkers to be associated with at least one of 433 sentinel variants across 247 regions (Fig. 2a, Supplementary Table 9, Supplementary Information). Ninety-five of the regions were associated with five or more biomarker categories (Extended Data Fig. 6c, Supplementary Table 9), including pleiotropic disease loci such as *MHC*, *APOE*, *GCKR*, *SH2B3* and *FUT2*.

### SLC34A1 deletion and fluid biomarkers.

Cross-referencing disease with biomarker associations provided mechanistic insights into novel findings. For instance, a low-frequency in-frame deletion in *SLC34A1* (rs1460573878; MAF = 2.6% (UKB) and 2.7% (FG); p.Val91_Ala97del) coding for the sodium phosphate cotransporter NPT2a expressed in proximal tubular cells was associated with increased risk of renal (log(OR) = 0.24, $P = 4.0 \times 10^{-9}$) and urinary tract stones (log(OR) = 0.21, $P = 6.8 \times 10^{-9}$). The deletion has previously been implicated in hypercalciuric renal stones[15,16] and autosomal recessive idiopathic infantile hypercalcaemia[17] in family studies. The variant is also associated with increased serum calcium ($\beta = 0.047$, $P = 5.4 \times 10^{-11}$) and reduced phosphate ($\beta = −0.075$, $P = 3.3 \times 10^{-26}$), consistent with a disrupted function or cell surface expression of the transporter[17] (Fig. 2b). We further find associations with increased levels of serum urate ($\beta = 0.048$, $P = 4.5 \times 10^{-17}$), also suggesting an increased risk of uric acid stones. Additionally, we found associations with increased erythrocyte count ($\beta = 0.035$, $P = 4.7 \times 10^{-10}$),

haemoglobin concentration ($\beta = 0.033$, $P = 7.7 \times 10^{-10}$) and haematocrit percentage ($\beta = 0.036$, $P = 9.9 \times 10^{-11}$), suggesting increased renal-driven erythropoiesis (Fig. 2b). Serum creatinine was not increased in carriers of the deletion ($\beta = −0.07$, $P = 3.6 \times 10^{-33}$), suggesting that renal function is not adversely affected in deletion carriers. Among 11,114 renal or ureteric, and 13,319 urinary tract stone cases, we identified 735 (renal or ureteric) and 863 (urinary tract) carriers of the deletion who may benefit from clinical interventions targeting NPT2A-related pathways and monitoring for disturbed biochemical and haematological biomarkers.

### CHEK2 deletion and haematological signs.

A frameshift deletion in *CHEK2* (rs555607708; MAF = 0.64% (FG), 0.24% (UKB)) that increases breast cancer risk has also been previously implicated in myeloproliferative neoplasms through GWAS[18] and lymphoid leukaemia in a candidate variant study[19]. Consistently, we found nominally significant associations with risks of both, myeloid (log(OR) = 1.52, $P = 9.5 \times 10^{-8}$) and lymphoid (log(OR) = 1.38, $P = 3.1 \times 10^{-7}$) leukaemia, but also multiple myeloma (log(OR) = 1.07, $P = 5.1 \times 10^{-5}$) and non-Hodgkin lymphoma (log(OR) = 0.81, $P = 4.7 \times 10^{-4}$). Association of rs555607708 with clinical haematology traits showed statistically significant associations with increased blood cell counts for both myeloid (leukocytes, neutrophils and platelets at $P < 1 \times 10^{-6}$; monocyte and erythrocytes at $P < 1 \times 10^{-3}$) and lymphoid (lymphocytes, $P = 5.7 \times 10^{-17}$) lineages (Fig. 2c). Furthermore, we found associations with increased mean platelet volume ($P = 1.3 \times 10^{-16}$) and platelet distribution width ($P = 5.2 \times 10^{-13}$), consistent with increased platelet activation and previous associations of mean platelet volume and platelet distribution width with chronic myeloid leukaemia[20]. We also found associations with decreased mean corpuscular haemoglobin ($P = 7.8 \times 10^{-12}$) and mean corpuscular volume ($P = 5.3 \times 10^{-10}$), suggesting that predisposition to haematological cancers by loss of *CHEK2* function is accompanied by a microcytic red blood cell phenotype (Fig. 2c).

### Coding associations aid drug development

We cross-referenced genes with significant coding variant associations with drug targets[21]. We found 66 genes with trait cluster associations that are the targets of either approved drugs (26 genes) or drugs currently being tested in clinical trials (40 genes), 14 of which are in phase III trials (Supplementary Table 10). We found a statistically significant enrichment of significant genes in our study that were also approved drug targets (26 out of 482, compared with a background of 569 approved targets out of 19,955 genes, OR = 1.9, $P = 0.0024$), which is in line with previous estimates of a higher success rate for drug targets supported by genetics[22,23]. Sensitivity analyses using more stringent association $P$-value thresholds further increased these probability estimates ($P = 5 \times 10^{-9}$ (OR = 2.3, $P = 0.00070$); $P = 5 \times 10^{-10}$ (OR = 2.5, $P = 0.00037$)), supporting previous observations of higher likelihood of therapeutic success with stronger genetic associations (Supplementary Table 11). Specific examples are highlighted in the Supplementary Information.

### Atrial fibrillation (AF).

GWAS have yielded a sizeable number of loci[24,25]. We chose AF to exemplify how results from our study can further elucidate the genetics and biological basis of one distinct human trait. Notably, we report several coding variant associations (Supplementary Table 3) in which prior GWAS[24,25] had fallen short for resolving GWAS loci to coding genes and explaining disease mechanisms.

### METTL11B methylase missense variant in AF.

The AF GWAS sentinel variant rs72700114 is an intergenic variant located between *METTL11B* and *LINCO1142* with no obvious candidate gene[24–26]. Our study revealed that a low-frequency missense variant in the methylase *METTL11B* (rs41272485:G; Ile127Met; MAF = 3.9% (UKB) and 3.8% (FG)) was associated with increased AF risk (log(OR) = 0.14, $P = 4.0 \times 10^{-11}$). This variant locates to the enzyme's *S*-adenosylmethionine −*S*-adenosyl-L-homocysteine ligand-binding site[27] and is expected to

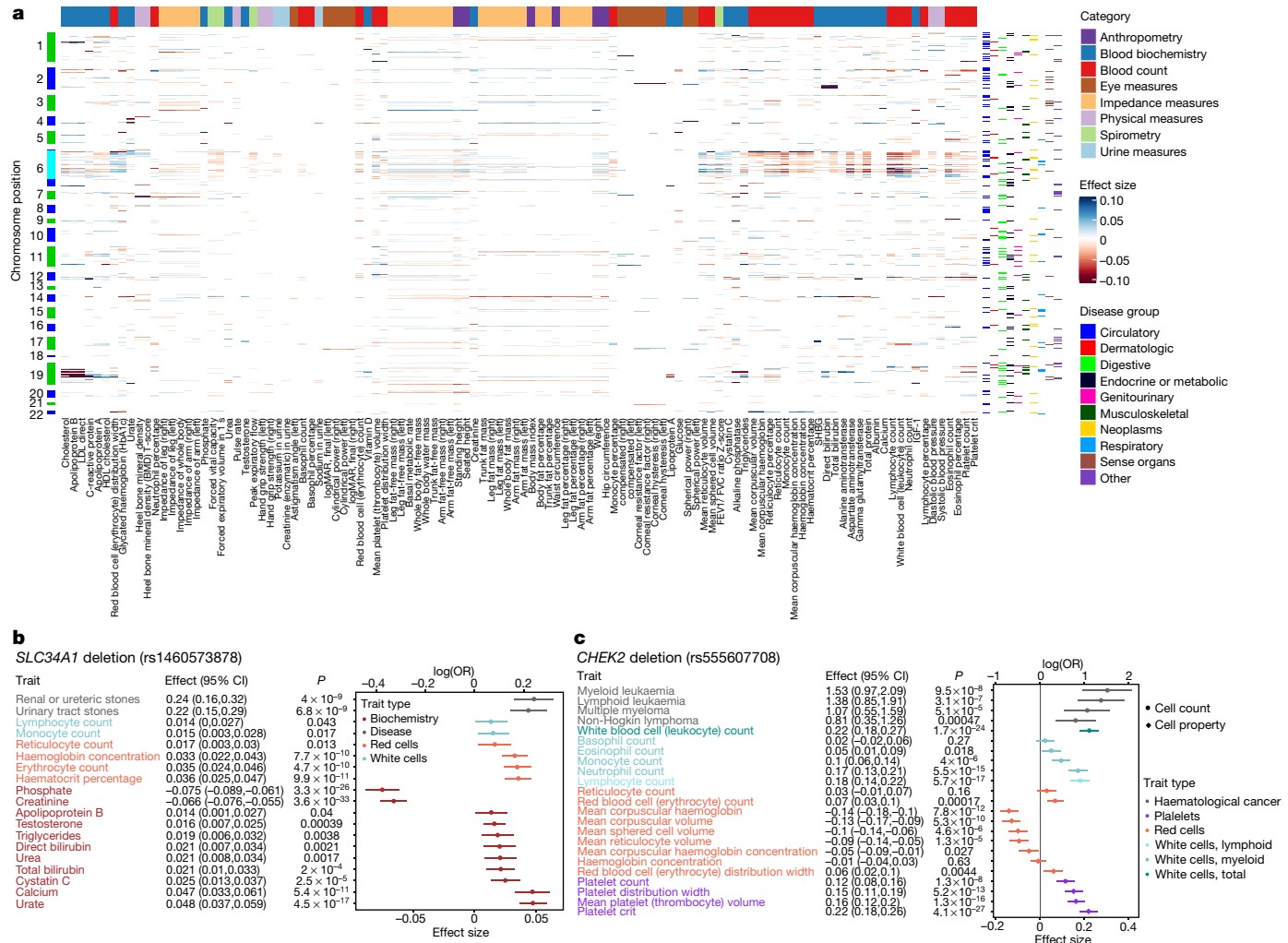

**Fig. 2 | Biomarker associations with sentinel variants. a**, Heat map of sentinel associations with biomarkers. Only significant associations ($P < 10^{-6}$) are shown. Colours on the left axis indicate chromosomes, with cyan indicating the MHC region. Colours on the right axis indicate sentinel association with disease by group. Colours along the top indicate the category of biomarkers. **b**, Forest plot of associations (unadjusted regression effect estimates with 95% confidence intervals (CI)) between *SLC34A1* deletion (rs1460573878) with haematological and biochemistry biomarkers. associations with $P < 0.05$ are shown. **c**, Forest plot of associations (unadjusted regression effect estimate with 95% confidence interval (CI)) between *CHEK2* deletion (rs555607708) with haematological biomarkers. Unadjusted *P* values are shown. Disease associations $n = 653,219$ biologically independent samples. Specific sample sizes for biomarker associations are listed in Supplementary Table 8. IGF-1, insulin-like growth factor 1; LDL, low-density lipoprotein; SHBG, sex hormone binding globulin.

perturb methylation of other AF risk genes with N-terminal (Ala/Pro/Ser)-Pro-Lys methylation motifs that are enriched in cardiomyocytes (Methods, Supplementary Table 12, Supplementary Results), which probably explains the association.

**Rare variants and ion channel AF loci.** Within the *SCN5A–SCN10A* locus, we replicated a common missense variant in *SCN10A* (rs6795970:A; Ala1073Val; MAF = 40.0% (UKB) and 44.6% (FG)) that was previously described as prolonging cardiac conduction[28]. Additionally, we found associations with reduced AF risk (log(OR) = −0.06, $P = 2.1 \times 10^{-12}$), reduced pulse rate ($\beta = -0.02$, $P = 4.8 \times 10^{-18}$), and a suggestive signal for increased risk of atrioventricular block (log(OR) = 0.10, $P = 1.9 \times 10^{-7}$). It is thus tempting to speculate that loss of function of Na$_V$1.8—the sodium channel encoded by *SCN10A*—blunts the effects of vagus nerve activity on the atria. In addition, we found a rare, enriched missense variant in FG (rs45620037:A; Thr220Ile; MAF = 0.11% (UKB) and 0.47% (FG); SIFT = 0.03, PolyPhen = 0.96) in *SCN5A*—which encodes the cardiac sodium channel Na$_V$1.5—to be associated with decreased risk of AF (log(OR) = −0.65, $P = 1.3 \times 10^{-12}$). This missense variant resides within

the voltage sensing segment of SCN5A, causes a partial loss of function of the Na$_V$1.5 channel in atrial cells and has been associated with dilated cardiomyopathy[29] and conduction defects including sick sinus syndrome and atrial standstill[30] in family studies with bradycardic changes. Consistently, we found a nominal association with reduced pulse rate ($\beta = -0.078$, $P = 0.023$), suggesting that protective effects of the variant will be most beneficial for the common tachycardic form of AF through reducing pulse rate. The *SCN10A* and *SCN5A* variants found here are probably both moderators of AF risk that act by different mechanisms. Potential mechanisms underlying further AF loci are described in Supplementary Discussion.

**Genetic effects underlying AF and pulse.** To further evaluate the hypothesis that distinct genetic mechanisms underlying AF risk inversely modulate pulse rate, we adjusted the clustered Mendelian randomization[31] (MR-Clust) algorithm to better accommodate rare-variants. We then related expectation maximization clustering of AF associated variants with homogenous directional effects on pulse rate (Methods). We found clusters of CWAS AF sentinel variants in *SCN10A* (rs6795970)

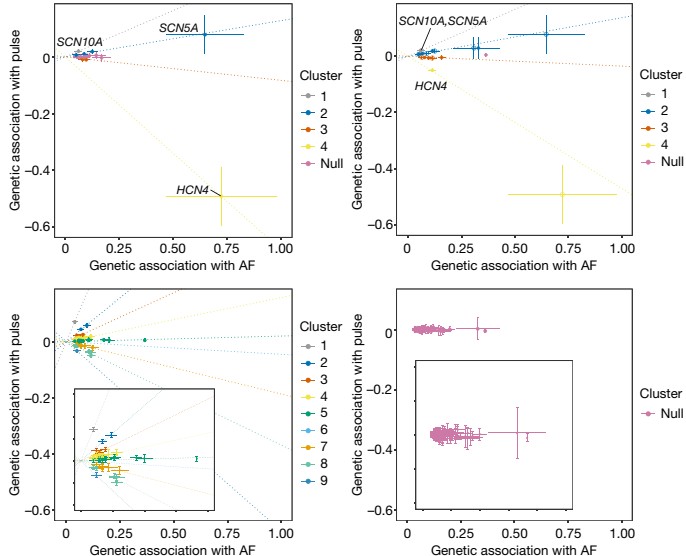

**Fig. 3 | Genetic and functional insights into atrial fibrillation.** Clustered Mendelian randomization plot of association of atrial fibrillation loci with pulse rate. Only variants with cluster inclusion probability greater than 0.7 are included. Top left, CWAS loci (sentinels). Top right, overlapping CWAS and atrial fibrillation GWAS loci. Bottom left, all atrial fibrillation GWAS loci from Nielsen et al.[49] (with zoomed inset). Bottom right, all atrial fibrillation GWAS loci with permuted pulse (null, with zoomed inset).

and *HCN4* (rs151004999) as two genetic components of AF that can increase and decrease pulse rate, respectively (Fig. 3, Supplementary Table 13). Identifying components of AF with diverging directionality on pulse rate matches clinical observations that AF can be caused by both tachycardia and bradycardia[32]. Using sentinel variants from a recent AF GWAS[24] for sensitivity analyses yielded concordant patterns. By integrating CWAS and GWAS sentinel variants for AF we found additional clusters with differing effects on pulse rate. Expectedly, within the AF GWAS loci[24], the two rare missense alleles in *HCN4* (rs151004999:T, log(OR) = 0.72) and *SCN5A* (rs45620037:A, log(OR) = −0.65) identified in our study had much larger effect sizes on AF risk than the respective non-coding sentinel GWAS variants (rs74022964:T (*HCN4* locus), log(OR) = 0.12; rs6790396:C (*SCN5A* and *SCN10A* locus), log(OR) = −0.058) (Fig. 3).

**Functional effects of PITX2c(Pro41Ser).** Finally, we found a rare missense variant in *PITX2* as associated with increased risk of AF (log(OR) = 0.38, $P = 1.1 \times 10^{-9}$). This variant is enriched nearly 50-fold in FG (rs143452464:A; Pro41Ser; MAF = 0.023% (UKB) and 1.1% (FG)) and was independently identified in a French family with AF (Supplementary Information), whereas GWAS had linked intergenic variants between *PITX2* and *FAM241A* to AF risk. PITX2 is a bicoid type homeobox transcription factor previously assumed to play a role in cardiac rhythm control[33]. The Pro41Ser variant lies in the N-terminal domain that is only present in the PITX2c isoform expressed in cardiac muscle. In reporter assays comparing the ability of PITX2c wild-type and Pro41Ser protein constructs to transactivate a luciferase reporter plasmid containing a putative PITX2c-binding element, PITX2c(Pro41Ser) showed an approximately 2.4-fold higher activation of the reporter than the wild type (*P* = 0.006, Extended Data Fig. 8). This effect was abrogated upon deletion of the putative PITX2c-binding site. In cultured cardiac muscle HL-1 cells, the Pro41Ser mutation increased the transcription of several presumed PITX2c target genes (Supplementary Table 14, Supplementary Information). Together, these results are consistent with a putative gain-of-function mechanism of Pro41Ser on PITX2c transactivation potential and AF risk.

## Discussion

Here we have conducted the largest association study of protein-coding genetic variants so far against hundreds of disease endpoints ascertained from two massive population biobanks, UK Biobank and FinnGen. We report novel disease associations, most notably in the rare and low-allelic frequency spectrum, replicate and assign putative causalities to many previously reported GWAS associations, and leverage the insights gained to elucidate disease mechanisms, demonstrating that the step from association to biological insight may be considerably shorter for coding variant association studies than it has traditionally been for GWAS. In addition to a substantial gain in power over previous studies, our analyses benefit from replication between two population cohorts, increasing the robustness of our findings and setting the stage for future similar studies in ethnically more diverse populations.

Notably, our study identifies both pathogenic variants residing in monogenic disease genes to impact the risk for related complex conditions as well as new, probably causal sentinel variants within GWAS loci in genes with known and novel biological roles in the respective GWAS trait. With this, our study is one of the first to help bridge the gap between common and rare disease genetics across a broad range of conditions and provides support for the hypothesis that the genetic architecture of many diseases is continuous[1]. Of the 975 associations identified in our study, 145 are driven by unique variants in the so far little-interrogated rare and low-allelic frequency spectrum between 0.1 and 2% that neither GWAS nor sequencing studies have been able to thoroughly interrogate across a range of diseases and that is hypothesized to contribute to the 'missing heritability' of many human diseases[34].

Our approach benefits considerably from the Finnish genetic background, where certain alleles are stochastically enriched to unusually high allele frequencies[6–8], at times exceeding population frequencies in the UK Biobank by more than 50-fold. Our theoretical and empirical results suggest the increasing utility of enriched variants for identifying associations quantitatively towards lower allelic frequencies. Notably, we identify the most prominent relative power gain in the rarest variant frequency spectrum, highlighting a role for sequencing studies and integrating additional population cohorts with enriched variants for identifying novel disease associations at scale. We identify several alleles with comparatively high effect sizes and a prevalence in the population that warrants follow-up, both experimentally as well as potentially directly in clinical settings to help improve disease outcomes. For instance, our data propose that 6.5% of UKB and FG participants with kidney or urinary tract stones, conditions debilitating more than 15% of men and 5% of women by 70 years of age[35], carry a deletion in *SLC34A1*. Monitoring patients for the clinical biomarkers identified here as associated with this deletion might help to differentiate aetiologies and guide individualized treatments. Similarly, coding variant associations identified in our study may serve as an attractive source to generate hypotheses for drug discovery programs. Our results support previous studies[22,23] that drug targets supported by human genetics have an increased likelihood of success, which can be considered particularly high when the genetic effect on a drug target closely mimics that of a pharmacological intervention[36].

Our results foreshadow the discovery of many additional coding and non-coding associations from cross-biobank analyses at even larger sample sizes. With the continued growth of population biobanks with comprehensive health data in non-European populations, the emergence of more and more cost-effective technologies for sequencing and genotyping, and computational advances to analyse genetic and non-genetic data at scale, future studies will be able to assess the genetic contribution to health and disease at even finer resolution.

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

**Biogen Biobank Team**

Benjamin B. Sun[1,2], Chia-Yen Ghen[1], Eric Marshall[1], Jemma B. Wilk[1] & Heiko Runz[1]

A full list of members and their affiliations appears in the Supplementary Information

**Finn Gen**

Mitja I. Kurki[3,4,5,6], Aarno Palotie[3,4,5,6] & Mark J. Daly[3,4,5,6]

A full list of members and their affiliations appears in the Supplementary Information

## Methods

### Samples and participants

UKB is a UK population study of approximately 500,000 participants aged 40–69 years at recruitment[2]. Participant data (with informed consent) include genomic, electronic health record linkage, blood, urine and infection biomarkers, physical and anthropometric measurements, imaging data and various other intermediate phenotypes that are constantly being updated. Further details are available at https://biobank.ndph.ox.ac.uk/showcase/. Analyses in this study were conducted under UK Biobank Approved Project number 26041. Ethic protocols are provided by the UK Biobank Ethics Advisory Committee (https://www.ukbiobank.ac.uk/learn-more-about-uk-biobank/about-us/ethics).

FG is a public-private partnership project combining electronic health record and registry data from six regional and three Finnish biobanks. Participant data (with informed consent) include genomics and health records linked to disease endpoints. Further details are available at https://www.finngen.fi/. More details on FG and ethics protocols are provided in Supplementary Information. We used data from FG participants with completed genetic measurements (R5 data release) and imputation (R6 data release). FinnGen participants provided informed consent for biobank research. Recruitment protocols followed the biobank protocols approved by Fimea, the National Supervisory Authority for Welfare and Health. The Coordinating Ethics Committee of the Hospital District of Helsinki and Uusimaa (HUS) approved the FinnGen study protocol Nr HUS/990/2017. The FinnGen study is approved by Finnish Institute for Health and Welfare.

### Disease phenotypes

FG phenotypes were automatically mapped to those used in the Pan UKBB (https://pan.ukbb.broadinstitute.org/) project. Pan UKBB phenotypes are a combination of Phecodes[37] and ICD10 codes. Phecodes were translated to ICD10 (https://phewascatalog.org/phecodes_icd10, v.2.1) and mapping was based on ICD-10 definitions for FG endpoints obtained from cause of death, hospital discharge and cancer registries. For disease definition consistency, we reproduced the same Phecode maps using the same ICD-10 definitions in UKB. In particular, we expertly curated 15 neurological phenotypes using ICD10 codes. We retained phenotypes where the similarity score (Jaccard index: $ICD10_{FG} \cap ICD10_{UKB} / ICD10_{FG} \cup ICD10_{UKB}$) was >0.7 and additionally excluded spontaneous deliveries and abortions.

Phecodes and ICD10 coded phenotypes were first mapped to unified disease names and disease groups using mappings from Phecode, PheWAS and icd R packages followed by manual curation of unmapped traits and diseases groups, mismatched and duplicate entries. Disease endpoints were mapped to Experimental Factor Ontology (EFO) terms using mappings from EMBL-EBI and Open Targets based on exact disease entry matches followed by manual curation of unmapped traits.

Disease trait clusters were determined through first calculating the phenotypic similarity via the cosine similarity, then determining clusters via hierarchical clustering on the distance matrix (1-similarity) using the Ward algorithm and cutting the hierarchical tree, after inspection, at height 0.8 to provide the most semantically meaningful clusters.

### Genetic data processing

**UKB genetic QC.** UKB genotyping and imputation were performed as described previously[2]. Whole-exome sequencing data for UKB participants were generated at the Regeneron Genetics Center (RGC) as part of a collaboration between AbbVie, Alnylam Pharmaceuticals, AstraZeneca, Biogen, Bristol-Myers Squibb, Pfizer, Regeneron and Takeda with the UK Biobank. Whole-exome sequencing data were processed using the RGC SBP pipeline as described[3,38]. RGC generated a QC-passing 'Goldilocks' set of genetic variants from a total of 454,803 sequenced UK Biobank participants for analysis. Additional quality control (QC) steps were performed prior to association analyses as detailed below.

**FG genetic QC.** Samples were genotyped with Illumina and Affymetrix arrays (Thermo Fisher Scientific). Genotype calls were made with GenCall and zCall algorithms for Illumina and AxiomGT1 algorithm for Affymetrix data. Sample, genotyping as well as imputation procedures and QC are detailed in Supplementary Information.

**Coding variant selection.** GnomAD v.2.0 variant annotations were used for FinnGen variants[39]. The following gnomAD annotation categories are included: pLOF, low-confidence loss-of-function (LC), in-frame insertion–deletion, missense, start lost, stop lost, stop gained. Variants have been filtered to imputation INFO score > 0.6. Additional variant annotations were performed using variant effect predictor (VEP)[40] with SIFT and PolyPhen scores averaged across the canonical annotations.

### Disease endpoint association analyses

For optimized meta-analyses with FG, analyses in UKB were performed in the subset of exome-sequence UKB participants with white European ancestry for consistency with FG ($n = 392,814$). We used REGENIE v1.0.6.7 for association analyses via a two-step procedure as detailed in ref. [41]. In brief, the first step fits a whole genome regression model for individual trait predictions based on genetic data using the leave one chromosome out (LOCO) scheme. We used a set of high-quality genotyped variants: MAF > 5%, MAC > 100, genotyping rate >99%, Hardy–Weinberg equilibrium (HWE) test $p > 10^{-15}$, <5% missingness and linkage-disequilibrium pruning (1,000 variant windows, 100 sliding windows and $r^2 < 0.8$). Traits where the step 1 regression failed to converge due to case imbalances were subsequently excluded from subsequent analyses. The LOCO phenotypic predictions were used as offsets in step 2 which performs variant association analyses using the approximate Firth regression detailed in ref. [41] when the $P$ value from the standard logistic regression score test is below 0.01. Standard errors were computed from the effect size estimate and the likelihood ratio test $P$-value. To avoid issues related to severe case imbalance and extremely rare variants, we limited association test to phenotypes with >100 cases and for variants with MAC ≥ 5 in total samples and MAC ≥ 3 in cases and controls. The number of variants used for analyses varies for different diseases as a result of the MAC cut-off for different disease prevalence. The association models in both steps also included the following covariates: age, age[2], sex, age*sex, age[2]*sex, first 10 genetic principal components (PCs).

Association analyses in FG were performed using mixed model logistic regression method SAIGE v0.39[42]. Age, sex, 10 PCs and genotyping batches were used as covariates. For null model computation for each endpoint each genotyping batch was included as a covariate for an endpoint if there were at least 10 cases and 10 controls in that batch to avoid convergence issues. One genotyping batch need be excluded from covariates to not have them saturated. We excluded Thermo Fisher batch 16 as it was not enriched for any particular endpoints. For calculating the genetic relationship matrix, only variants imputed with an INFO score >0.95 in all batches were used. Variants with >3% missing genotypes were excluded as well as variants with MAF < 1%. The remaining variants were linkage-disequilibrium pruned with a 1-Mb window and $r^2$ threshold of 0.1. This resulted in a set of 59,037 well-imputed not rare variants for GRM calculation. SAIGE options for null computation were: "LOCO=false, numMarkers=30, traceCVcutoff=0.0025, ratioCVcutoff=0.001". Association tests were performed phenotypes with case counts >100 and for variants with minimum allele count of 3 and imputation INFO >0.6 were used.

We additionally performed sex-specific associations for a subset of gender-specific diseases (60 female diseases and in 50 disease clusters, 14 male diseases and in 13 disease clusters) in both FG and UKB using the same approach without inclusion of sex-related covariates (Supplementary Table 2)

We performed fixed-effect inverse-variance meta-analysis combining summary effect sizes and standard errors for overlapping variants with matched alleles across FG and UKB using METAL[43].

## Definition and refinement of significant regions

To define significance, we used a combination of (1) multiple testing corrected threshold of $P < 2 \times 10^{-9}$ (that is, $0.05/$(approximately $26.8 \times 10^6$), the sum of the mean number of variants tested per disease cluster)), to account for the fact that some traits are highly correlated disease subtypes, (2) concordant direction of effect between UKB and FG associations, and (3) $P < 0.05$ in both UKB and FG.

We defined independent trait associations through linkage-disequilibrium-based ($r^2 = 0.1$) clumping ±500 kb around the lead variants using PLINK[44], excluding the HLA region (chr6:25.5-34.0Mb) which is treated as one region due to complex and extensive linkage-disequilibrium patterns. We then merged overlapping independent regions (±500 kb) and further restricted each independent variant ($r^2 = 0.1$) to the most significant sentinel variant for each unique gene. For overlapping genetic regions that are associated with multiple disease endpoints (pleiotropy), to be conservative in reporting the number of associations we merged the overlapping (independent) regions to form a single distinct region (indexed by the region ID column in Supplementary Table 3).

## Cross-reference with known associations

We cross-referenced the sentinel variants and their proxies ($r^2 > 0.2$) for significant associations ($P < 5 \times 10^{-8}$) of mapped EFO terms and their descendants in GWAS Catalog[11] and PhenoScanner[12]. To be more conservative with reporting of novel associations, we also considered whether the most-severe associated gene in our analyses were reported in GWAS Catalog and PhenoScanner. In addition, we also queried our sentinel variants in ClinVar[13] to define known associations with rarer genetic diseases and further manually curated novel associations (where the association is a novel variant association and a novel gene association) for previous genome-wide significant ($P < 5 \times 10^{-8}$) associations.

To assess medical actionability of associated genes, we cross-referenced the associated genes with the latest ACMG v3. (75 unique genes linked to 82 conditions, linked to cancer ($n = 28$), cardiovascular ($n = 34$), metabolic ($n = 3$), or miscellaneous conditions ($n = 8$)). This list was supplemented by 20 'ACMG watchlist genes'[14] for which evidence for inclusion to ACMG 3.0 list was considered too preliminary based on either technical, penetrance or clinical management concerns

## Biomarker associations of lead variants

For the lead sentinel variants, we performed association analyses using the two-step REGENIE approach described above with 117 biomarkers including anthropometric traits, physical measurements, clinical haematology measurements, blood and urine biomarkers available in UKB (detailed in Supplementary Table 8). Additional biochemistry subgroupings were based on UKB biochemistry subcategories: https://www.ukbiobank.ac.uk/media/oiudpjqa/bcm023_ukb_biomarker_panel_website_v1-0-aug-2015-edit-2018.pdf

## Drug target mapping and enrichment

We mapped the annotated gene for each sentinel variant to drugs using the therapeutic target database (TTD)[21]. We retained only drugs which have been approved or are in clinical trial stages. For enrichment analysis of approved drugs with genetic associations, we used Fisher's exact test on the proportion of significant genes targeted by approved drug against a background of all approved drugs in TTD[21] ($n = 595$) and 20,437 protein coding genes from Ensembl annotations[45].

## Mendelian randomization analyses

*F5* and *F10* effects on pulmonary embolism. The missense variants rs4525 and rs61753266 in *F5* and *F10* genes were taken as genetic instruments for Mendelian randomization analyses. To assess potential that each factor level is causally associated with pulmonary embolism we used two-sample Mendelian randomization using summary statistics, with effect of the variants on their respective factor levels obtained from previous large scale (protein quantitative trait loci) pQTL studies[46,47]. Let $\beta_{XY}$ denote the estimated causal effect of a factor level on pulmonary embolism risk and $\beta_X$, $\beta_Y$ be the genetic association with a factor level (FV, FX or FXa) and pulmonary embolism risk respectively. Then, the Mendelian randomization ratio-estimate of $\beta_{XY}$ is given by:

$$\beta_{XY} = \frac{\beta_Y}{\beta_X}$$

where the corresponding standard error se($\beta_{XY}$), computed to leading order, is:

$$se(\beta_{XY}) = \frac{se(\beta_Y)}{|\beta_X|}$$

**Clustered Mendelian randomization.** To assess evidence of several distinct causal mechanisms by which AF may influence pulse rate (PR) we used MR-Clust[31]. In brief, MR-Clust is a purpose-built clustering algorithm for use in univariate Mendelian randomization analyses. It extends the typical Mendelian randomization assumption that a risk factor can influence an outcome via a single causal mechanism[48] to a framework that allows one or more mechanisms to be detected. When a risk-factor affects an outcome via several mechanisms, the set of two-stage ratio-estimates can be divided into clusters, such that variants within each cluster have similar ratio-estimates. As shown in[31], two or more variants are members of the same cluster if and only if they affect the outcome via the same distinct causal pathway. Moreover, the estimated causal effect from a cluster is proportional to the total causal effect of the mechanism on the outcome. We included variants within clusters where the probability of inclusion >0.7. We used MR-Clust algorithm allowing for singletons/outlier variants to be identified as their own 'clusters' to reflect the large but biologically plausible effect sizes seen with rare and low-frequency variants.

## Bioinformatic analyses for *METTL11B*

We searched [Ala/Pro/Ser]-Pro-Lys motif containing proteins using the 'peptide search' function on UniProt[49], filtering for reviewed Swiss-Prot proteins and proteins listed in Human Protein Atlas[50] (HPA) ($n = 7,656$). We obtained genes with elevated expression in cardiomyocytes ($n = 880$) from HPA based on the criteria: 'cell_type_category_rna: cardiomyocytes; cell type enriched, group enriched, cell type enhanced' as defined by HPA at https://www.proteinatlas.org/humanproteome/celltype/Muscle+cells#cardiomyocytes (accessed 20th March 2021) with filtering for those with valid UniProt IDs (Swiss-Prot, $n = 863$). Enrichment test was performed using Fisher's exact test. Additionally, we performed enrichment analyses using any [Ala/Pro/Ser]-Pro-Lys motif positioned within the N-terminal half of the protein ($n = 4,786$).

**Additional methods** Additional methods on further FinnGen QC; theoretical description and simulation of the effect of MAF enrichment on inverse-variance weighted (IVW) meta-analysis $Z$-scores; and functional characterization of PITX2c(Pro41Ser) are provided in the Supplementary Information.

## Reporting summary

Further information on research design is available in the Nature Research Reporting Summary linked to this paper.

## Data availability

Full summary association results of this study are accessible at https://doi.org/10.5281/zenodo.5571000. Summary and individual-level whole-exome sequencing data from UKB participants have been

deposited with UKB and will be freely available to approved researchers via The UK Biobank Research Analysis Platform (https://www.ukbiobank.ac.uk/enable-your-research/research-analysis-platform). FG summary association results are being released bi-annually via https://www.finngen.fi/en/access_results and can be explored in a public results browser (https://r5.finngen.fi). All analyses in this manuscript which rely on variants that were directly interrogated through chip-based genotyping with the FG array rely on FG data freeze 5 (from 11 May 2021). Analyses in this manuscript that are based on imputed variants rely on FG data freeze 6, which is anticipated to become public in November 2021. Individual-level genotypes and register data from FG participants can be accessed by approved researchers via the Fingenious portal (https://site.fingenious.fi/en/) hosted by the Finnish Biobank Cooperative FinBB (https://finbb.fi/en/). Data release to FinBB is timed to the bi-annual public release of FG summary results which occurs twelve months after FG consortium members can start working with the data. Further datasets underlying this study have been derived from: Therapeutic Target Database (http://db.idrblab.net/ttd/); Phecode-ICD10 data (https://phewascatalog.org/phecodes_icd10); GWAS Catalog (https://www.ebi.ac.uk/gwas/); PhenoScanner (http://www.phenoscanner.medschl.cam.ac.uk/); ClinVar (https://www.ncbi.nlm.nih.gov/clinvar/); gnomAD (https://gnomad.broadinstitute.org/); Human Protein Atlas (https://www.proteinatlas.org/); and Ensembl (https://www.ensembl.org/index.html).

## Code availability

Custom analysis scripts used are available at https://github.com/cnfoley/Sun-et-al-2021-protein-coding-variants-in-human-disease.

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

**Acknowledgements** We thank all the participants, contributors and researchers of UK Biobank and FinnGen (and its participating biobanks) for making data available for this study. We thank the UK Biobank Exome Sequencing Consortium (AbbVie, Alnylam Pharmaceuticals, AstraZeneca, Biogen, Bristol-Myers Squibb, Pfizer, Regeneron and Takeda) for generation of the whole-exome sequencing data and Regeneron Genetics Centre for initial quality control of the exome sequencing data. The FinnGen project is funded by two grants from Business Finland (HUS 4685/31/2016 and UH 4386/31/2016) and the following industry partners: AbbVie, AstraZeneca UK, Biogen MA, Celgene, Celgene International II, Genentech, Merck Sharp & Dohme, Pfizer, GlaxoSmithKline Intellectual Property Development, Sanofi US Services, Maze Therapeutics, Janssen Biotech and Novartis. We thank S. Lemmelä for her contribution to FinnGen data curation; and Y.-Q. Yang, T. Footz, M. Walter, A. Aránega, F. Hernández-Torres, E. Morel and G. Millat for their contributions to the functional characterization of PITX2c. PITX2 functional work was supported in part by grants from the National Natural Science Fund of China (81070153), the Personnel Development Foundation of Shanghai, China (2010019) and the Key Program of Basic Research of Shanghai, China (10JC1414002), and by the Canadian Institutes of Health Research (grants MOP-111072 and MOP-130373 to M.C.). Asma Mechakra was supported by a bursary of the French Ministry of Research and Technology (MRT).

**Author contributions** Conceptualization and experimental design: B.B.S. and H.R. Methodology: B.B.S., H.R., C.N.F., C.-Y.C. and M.J.D. Analysis: B.B.S., M.I.K., C.N.F., A.M., C.-Y.C., E.M., J.B.W. and Biogen Biobank Team. Experimental work: A.M., G.C., M.C. and P.C. FinnGen protocols and analysis: M.I.K., A.P., M.J.D. and FinnGen. Writing: B.B.S. and H.R. All authors critically reviewed the manuscript.

**Competing interests** B.B.S., H.R., C.-Y.C., E.M., J.W. and members of the Biogen Biobank Team are employees of Biogen. M.J.D. is a founder of Maze Therapeutics. The other authors declare no competing interests.

**Additional information**
**Correspondence and requests for materials** should be addressed to Benjamin B. Sun or Heiko Runz.

#1R5 of chip-based genotyping data (n=150,831), R6 of imputed data (n=260,405)
#2only included matching (score≥0.7) endpoints with cases ≥ 100
#3variants vary after filtering for MAC≥5 and MAC≥3 in cases

**Extended Data Fig. 1 | UKB and FG study overview.**

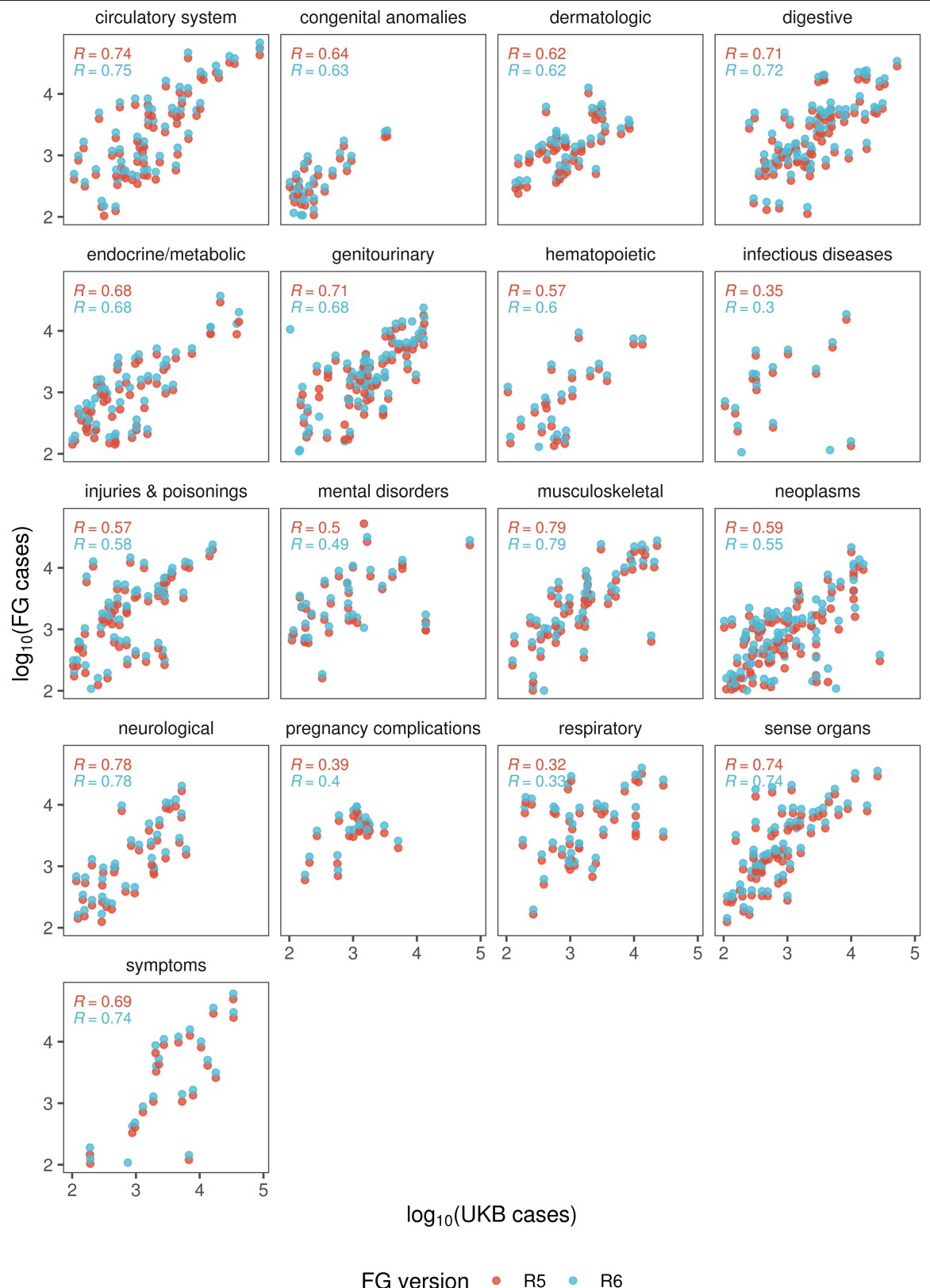

**Extended Data Fig. 2 | Case count comparison between UKB and FG across disease groups.** Diseases within each group are listed in Supplementary Table 2. Only cases >100 in UKB/FG are included. *R: Spearman's correlation for FG R5 (red) and R6 (blue).*

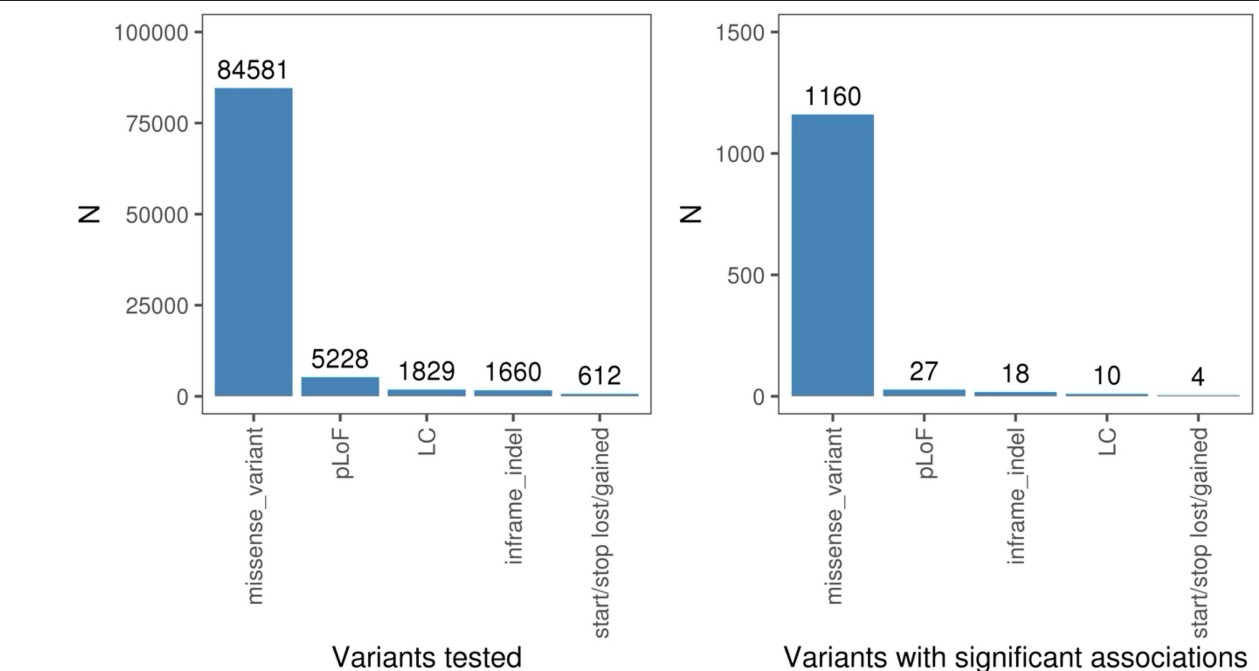

**Extended Data Fig. 3 | Distribution of variant annotation categories. Left**: all variants tested. **Right**: variants with at least 1 significant association ($p < 5 \times 10^{-8}$). pLOF: predicted loss of function. LC: low confidence loss of function.

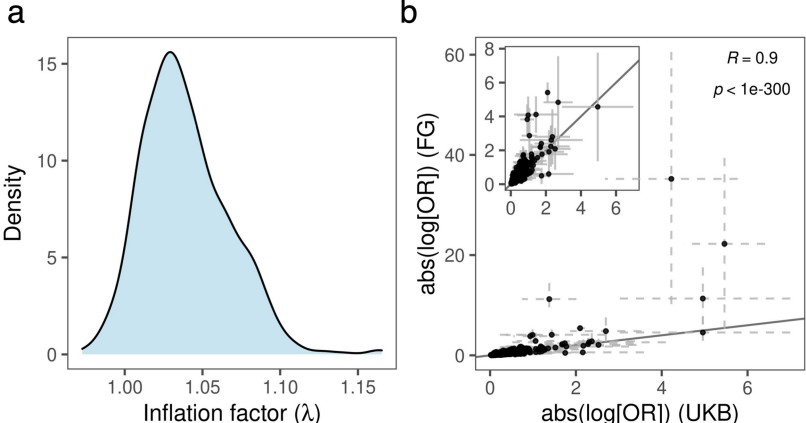

**Extended Data Fig. 4 | Inflation factors and FG-UKB effect size comparisons. (a)** Distribution of inflation factors of CWAS meta-analysis. (**b**) Effect size comparison between UKB and FG. Inset: zoomed in on small effect sizes. *R: Spearman's rank correlation (two-sided test), p* = 4.4 × 10$^{-351}$.

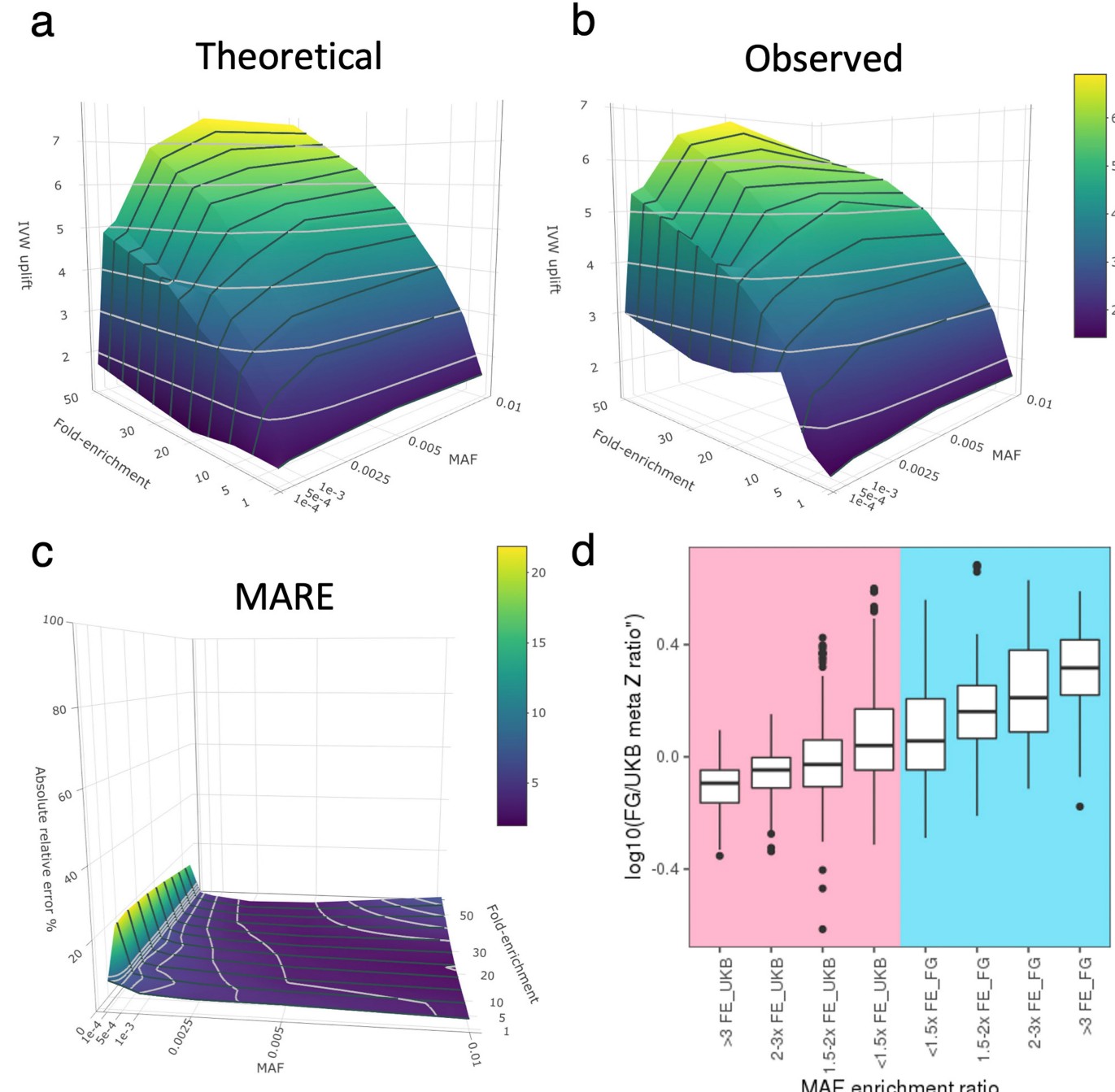

**Extended Data Fig. 5 | Surface plot of effects of cohort specific allele enrichment on inverse variant weighted meta-analysis z-scores (IVW uplift) across MAFs (up to MAF 1%).** Uplift is defined as the ratio of meta-analysed IVW Z-score to the Z-score of an individual study. **(a)** Theoretically predicted IVW uplift. **(b)** Observed IVW uplift. **(c)** Median absolute relative error (MARE, %) between simulated and theoretical IVW uplift values. For each combination of MAF and allelic enrichment, we simulated 1000 datasets for two binary variables reflecting disease status for two studies. Study sample size and disease prevalence were fixed (matching values estimates from UKB and FG), genomic effects were randomly sampled from the set of positive effect sizes in UKB and FG (Supplementary Table 3), MAF was varied from 0.01% to 1% and allele enrichment (in the smaller study)

ranged from 1 to 50. **(d)** Comparison of Z-scores for randomly subsetted UKB data meta-analysed with FinnGen (UKBxFG) against subsetted UKB meta-analysed with sample size matched UKB cohort (UKBxUKB') across allele fold enrichments for sentinel associations (Supplementary Table 3) with MAF<0.1. Y-axis ($\log_{10}$(FG/UKB meta Z ratio): $\log_{10}(Z_{UKBxFG}/Z_{UKBxUKB'})$. X-axis (MAF enrichment ratio): allelic fold enrichment (FE) where pink side denotes greater enrichment in UKB, blue side denotes greater enrichment in FG. Each box plot presents the median, first and third quartiles, with upper and lower whiskers representing 1.5x inter-quartile range above and below the third and first quartiles respectively. N for each boxplot from left to right: 88, 99, 212, 453, 213, 71, 126, 158.

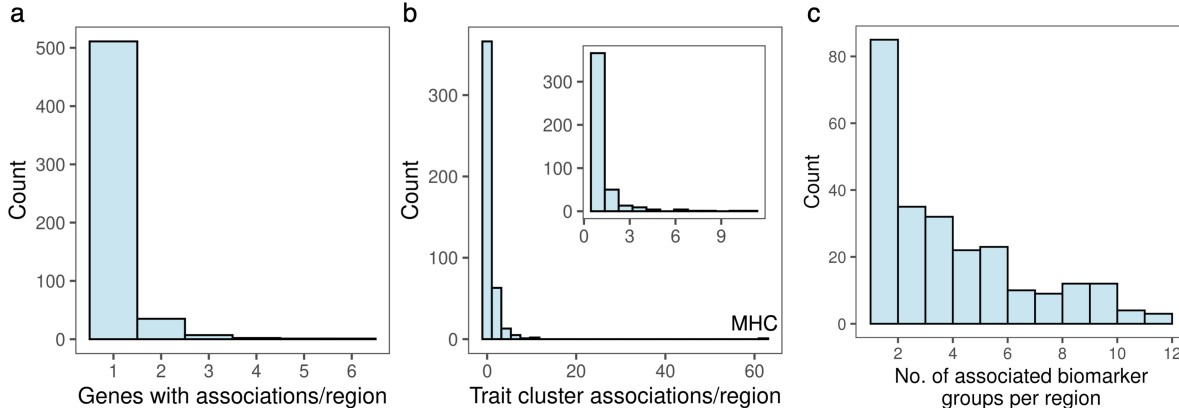

**Extended Data Fig. 6 | Histogram of genes with associations, disease and biomarker associations per region. (a)** Number of genes with coding associations per region. Each disease cluster counted separately. MHC region excluded. **(b)** Number of associated trait clusters ($p < 5 \times 10^{-8}$) per region. Inset shows zoomed in x-scale between 0-12 trait cluster associations per region. **(c)** Number of associated biomarker groups per locus ($p < 1 \times 10^{-6}$). *MHC: Major Histocompatibility Complex.*

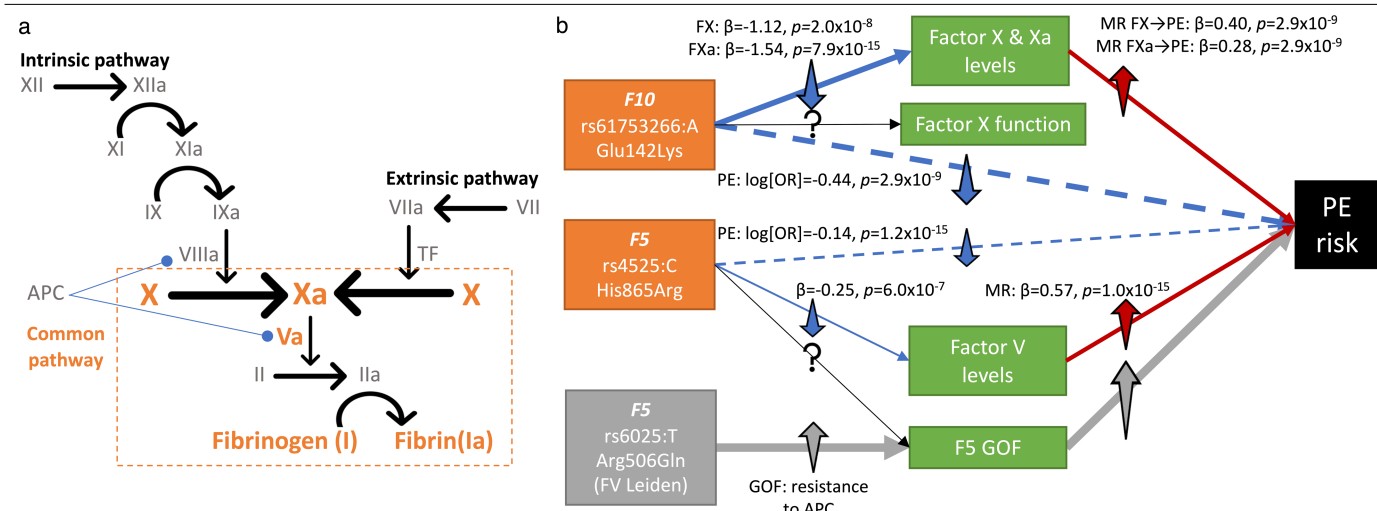

**Extended Data Fig. 7 | (a) Simplified diagram of the coagulation cascade.** Factors (in roman numerals, "a" represents activated) with genetic association with PE highlighted in orange. Blue line (round end) indicates inhibitory effect of APC on VIIIa and Va. (**b**) Schematic of potential pathway from missense variants in *F5* and *F10* to PE risk. Factor V Leiden variant had null associations with F5 levels ($\beta_{F5 levels}=0.21$, $p = 0.091$). Dashed blue lines suggest effect of the variants on PE risk which we assume under MR framework acts through factor levels (solid blue lines). Grey box and arrows represent known pathway for Factor V Leiden mutation. *GOF: Gain of function, APC: Activated protein C, MR: Mendelian randomisation, PE: Pulmonary embolism.*

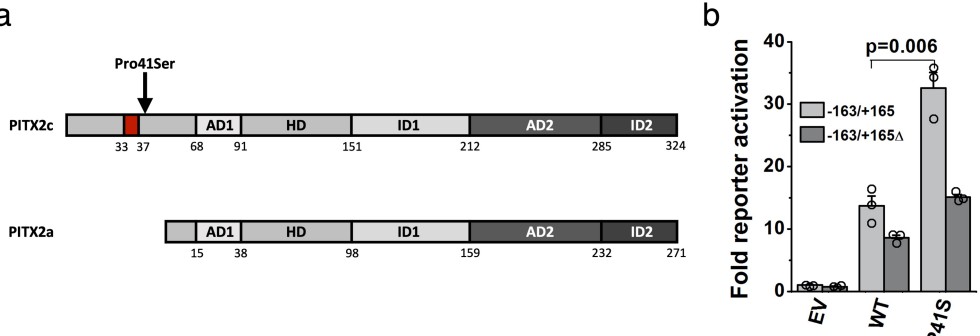

**Extended Data Fig. 8 | In vitro functional effects of the PITX2c Pro41Ser variant (rs143452464).** (**a**) Schematic of the location of the Pro41Ser variant in PITX2c as compared with the PITX2a splicing isoform. Numbers below each row indicate AA number from N-terminal (left) to C-terminal (right). AD1: common sequence, HD: homeodomain, ID1: transcriptional inhibitory domain 1, AD2: second common sequence, ID2: transcriptional inhibitory domain 2. Pro41Ser lies within the terminal domain (grey), near the 5-amino acid LAMAS (single amino acid code) sequence (33 to 37 red), which is important for transcriptional activity of the N-terminal of PITX2c. (**b**) Reporter gene assays: TM-1 cells (Transformed human trabecular meshwork cells) were co-transfected with a firefly luciferase reporter plasmid containing a PITX2c binding element, a β-galactosidase control vector and expression vector for PTIX2c (wild-type, Pro41Ser (P41S) or empty control vector (EV)). The activity of firefly luciferase upon activation by PITX2c (n = 3 transfections per condition) was normalized to β-galactosidase. Experiments with a truncated reporter construct ("−163/+165Δ"), containing a deletion of 8bp within the predicted PITX2 binding site, are shown as additional control. Data are presented as mean values +/− SEM, unadjusted *p*-value derived from two-sided t-test (EVΔ vs EV: 0.122; WTΔ vs WT: 0.074; P41SΔ vs P41S: 0.018; WT vs EV: 0.015; P41S vs WT: 0.0056; WTΔ vs EVΔ: 0.002; P41SΔ vs WTΔ: 0.0005).

# nature research

# Reporting Summary

Nature Research wishes to improve the reproducibility of the work that we publish. This form provides structure for consistency and transparency in reporting. For further information on Nature Research policies, see our Editorial Policies and the Editorial Policy Checklist.

## Statistics

For all statistical analyses, confirm that the following items are present in the figure legend, table legend, main text, or Methods section.

| n/a | Confirmed | |
|---|---|---|
| ☐ | ☒ | The exact sample size (*n*) for each experimental group/condition, given as a discrete number and unit of measurement |
| ☐ | ☒ | A statement on whether measurements were taken from distinct samples or whether the same sample was measured repeatedly |
| ☐ | ☒ | The statistical test(s) used AND whether they are one- or two-sided *Only common tests should be described solely by name; describe more complex techniques in the Methods section.* |
| ☐ | ☒ | A description of all covariates tested |
| ☐ | ☒ | A description of any assumptions or corrections, such as tests of normality and adjustment for multiple comparisons |
| ☐ | ☒ | A full description of the statistical parameters including central tendency (e.g. means) or other basic estimates (e.g. regression coefficient) AND variation (e.g. standard deviation) or associated estimates of uncertainty (e.g. confidence intervals) |
| ☐ | ☒ | For null hypothesis testing, the test statistic (e.g. *F*, *t*, *r*) with confidence intervals, effect sizes, degrees of freedom and *P* value noted *Give P values as exact values whenever suitable.* |
| ☒ | ☐ | For Bayesian analysis, information on the choice of priors and Markov chain Monte Carlo settings |
| ☒ | ☐ | For hierarchical and complex designs, identification of the appropriate level for tests and full reporting of outcomes |
| ☐ | ☒ | Estimates of effect sizes (e.g. Cohen's *d*, Pearson's *r*), indicating how they were calculated |

*Our web collection on statistics for biologists contains articles on many of the points above.*

## Software and code

Policy information about availability of computer code

| Data collection | No specific software were employed in collection |
|---|---|
| Data analysis | R v3.6.1 including packages: PheWAS v0.99.5, icd v4.0.9, MRClust (custom), RSpectra v0.16, GenomicRanges v1.38, ComplexHeatmap v2.7.11 REGENIE v1.0.6.7 SAIGE v0.39 METAL v2011-03-2 PLINK v1.9 and v2.0 AxiomGT1 Eagle v2.3.5 SISu v3 Hail v0.1 VEP v103 Details of specific software and references, including genetic measurements and QC, can be found within text in the relevant Methods and Supplementary Information sections. Code availability statement included. Code availability Custom analysis scripts used are available at https://github.com/cnfoley/Sun-et-al-2021-protein-coding-variants-in-human-disease. |

For manuscripts utilizing custom algorithms or software that are central to the research but not yet described in published literature, software must be made available to editors and reviewers. We strongly encourage code deposition in a community repository (e.g. GitHub). See the Nature Research guidelines for submitting code & software for further information.

## Data

Policy information about availability of data

All manuscripts must include a data availability statement. This statement should provide the following information, where applicable:
- Accession codes, unique identifiers, or web links for publicly available datasets
- A list of figures that have associated raw data
- A description of any restrictions on data availability

Data availability statement present and updated.
Data availability
Full summary association results of this study are accessible at https://doi.org/10.5281/zenodo.5571000. Summary and individual-level whole exome sequencing data from UKB participants have been deposited with UKB and will be freely available to approved researchers via The UK Biobank Research Analysis Platform (https://www.ukbiobank.ac.uk/enable-your-research/research-analysis-platform). FG summary association results are being released bi-annually via https://www.finngen.fi/en/access_results and can be explored in a public results browser (https://r5.finngen.fi). Individual-level genotypes from FinnGen participants can be accessed by approved researchers via the Fingenious portal (https://site.fingenious.fi/en/) hosted by the Finnish Biobank Cooperative FinBB (https://finbb.fi/en/). Further datasets underlying this study have been derived from: Therapeutic Target Database (http://db.idrblab.net/ttd/); Phecode-ICD10 data (https://phewascatalog.org/phecodes_icd10); GWAS Catalog (https://www.ebi.ac.uk/gwas/); PhenoScanner (http://www.phenoscanner.medschl.cam.ac.uk/); ClinVar (https://www.ncbi.nlm.nih.gov/clinvar/); gnomAD (https://gnomad.broadinstitute.org/); Human Protein Atlas (https://www.proteinatlas.org/); and Ensembl (https://www.ensembl.org/index.html).

# Field-specific reporting

Please select the one below that is the best fit for your research. If you are not sure, read the appropriate sections before making your selection.

☒ Life sciences  ☐ Behavioural & social sciences  ☐ Ecological, evolutionary & environmental sciences

For a reference copy of the document with all sections, see nature.com/documents/nr-reporting-summary-flat.pdf

# Life sciences study design

All studies must disclose on these points even when the disclosure is negative.

| Sample size | Methods, "Samples and participants"; Introduction sections. Note: this is a discovery study across multiple phenotypes and thus we did not perform specific power calculations to detect pre-specified effect sizes - but we are well-powered as our study combines two of the largest available population cohorts than previous studies. Sample size were chosen from the largest data available to maximise power, since this is a discovery study, the goal is to maximise power rather than to detect at specific effect sizes. |
|---|---|
| Data exclusions | Methods, "Genetic data processing", "Disease endpoint association analyses", Extended Data Figure 1. Samples failing QC and non-overlapping phenotypes were excluded as per methods. |
| Replication | Methods, "Definition and refinement of significant regions". In addition to multiple testing correction, we also required replication at p<0.05 in UKB/FG with concordant direction of effects. With in results, we also include cases where replication associations in previous GWAS. Replication/all summary data are available online. |
| Randomization | No experimental vs control group per se. |
| Blinding | No experimental vs control group per se. All data are anonymised and analysts were blind to sample statuses. |

# Reporting for specific materials, systems and methods

We require information from authors about some types of materials, experimental systems and methods used in many studies. Here, indicate whether each material, system or method listed is relevant to your study. If you are not sure if a list item applies to your research, read the appropriate section before selecting a response.

## Materials & experimental systems

| n/a | Involved in the study |
|---|---|
| ☐ | ☒ Antibodies |
| ☐ | ☒ Eukaryotic cell lines |
| ☒ | ☐ Palaeontology and archaeology |
| ☒ | ☐ Animals and other organisms |
| ☐ | ☒ Human research participants |
| ☒ | ☐ Clinical data |
| ☒ | ☐ Dual use research of concern |

## Methods

| n/a | Involved in the study |
|---|---|
| ☒ | ☐ ChIP-seq |
| ☒ | ☐ Flow cytometry |
| ☒ | ☐ MRI-based neuroimaging |

# Antibodies

| | |
|---|---|
| Antibodies used | For Western blots on HL-1 cell protein extracts, antibodies were a mouse monoclonal anti-V5 (V8012, Sigma-Aldrich) and a mouse monoclonal anti-GAPDH (G8795, Sigma-Aldrich). Batch numbers unknown.<br><br>Antibodies used with TM-1 cells experiments:<br>Anti-Xpress Monoclonal Antibody was purchased from ThermoFisher Scientific (#R910-25). Batch number unknown.<br><br>Goat Anti-Mouse-IgG coupled to Cyanine3: Cy™3 AffiniPure Goat Anti-Mouse IgG (H+L) polyclonal, was purchased from Jackson ImmunoResearch Laboratories Inc. (#115-165-003). Batch number unknown. (115-165-003). |
| Validation | V8012, Sigma-Aldrich<br>https://www.sigmaaldrich.com/FR/fr/search/v8012?focus=products&page=1&perPage=30&sort=relevance&term=V8012&type=product<br>the website says: V8012 V5-10, monoclonal WB, ICC<br>https://www.sigmaaldrich.com/FR/fr/search/v8012?focus=products&page=1&perPage=30&sort=relevance&term=V8012&type=product<br>the website says: V8012 V5-10, monoclonal WB, ICC<br>G8795, Sigma-Aldrich<br>https://www.sigmaaldrich.com/FR/fr/search/g8795?focus=products&page=1&perPage=30&sort=relevance&term=G8795&type=product<br>the website says: G8795 GAPDH-71.1, monoclonal WB, ARR, ICC, ELISA  mouse, mink, rabbit, rat, human, hamster, canine, turkey, chicken, monkey, bovine.<br>Sigma-Aldrich quote papers and show images from the literature in which Abs V8012 and G8795 were used, but do not endorse them as a validation.<br>R910-25, Thermofisher<br>https://www.thermofisher.com/antibody/product/Xpress-Antibody-Monoclonal/R910-25.<br>The website says: "R910-25 is tested in Western blot against 100 ng of an E. coli expressed fusion protein containing the Xpress epitope."<br>115-165-003, Jackson Immunoresearch<br>https://www.jacksonimmuno.com/catalog/products/115-165-003<br>the website does not provide validation. |

# Eukaryotic cell lines

Policy information about cell lines

| | |
|---|---|
| Cell line source(s) | The TM-1 cell line (immortalized human Trabecular Meshwork cells) was a gift from Dr. Vincent Raymond, and was cultured in Dulbecco's Modified Eagle's Medium - low glucose (Sigma, #D6046), see Filla et. al., 2002, IOVS. 43:151; PMID: 11773026. HL-1 cells were a gift from Dr. W.C. Claycomb (doi: 10.1073/pnas.95.6.2979)  and were cultured using his dedicated "Claycomb Medium" ordered from Sigma-Aldrich (product 51800C) and the Fetal Calf Serum lot certified by Dr. Claycomb, ordered from Sigma-Aldrich (product F2442 lot #058K8426). Since Dr. W.C. Claycomb passed-away the HL-1 cells are provided by SIGMA and other firms. |
| Authentication | TM-1 cells are not authenticated.<br>HL-1 cells were authenticated by Dr. W.C. Claycomb.<br>No authentication tests have been performed in our laboratories. |
| Mycoplasma contamination | No cell line was tested for mycoplasma contamination in our laboratories. |
| Commonly misidentified lines<br>(See ICLAC register) | The ISLAC register searches for TM-1 or HTM or HTMC and for HL-1 did not report any match. |

# Human research participants

Policy information about studies involving human research participants

| | |
|---|---|
| Population characteristics | Details can be found in Supplementary table 1 which contains UKB and FG population characteristics. Also described in Introduction and in Methods "Samples and participants" sections. UKB and FG have been commonly described in the public domain - they are two population cohorts from the UK and Finland respectively. |
| Recruitment | UKB and FG are population cohorts sampled from across UK sites and Finnish biobanks respectively. Detailed in Methods ("Samples and participants") and Supplementary Information. |
| Ethics oversight | Methods ("Samples and participants") and Supplementary Information for approval and ethics details. In detail:<br>Analyses in this study were conducted under UK Biobank Approved Project number 26041. UK Biobank has approval from the North West Multi-centre Research Ethics Committee (MREC), which covers the UK. It also sought the approval in England and Wales from the Patient Information Advisory Group (PIAG) for gaining access to information that would allow it to invite people to participate. PIAG has since been replaced by the National Information Governance Board for Health & Social Care (NIGB). In Scotland, UK Biobank has approval from the Community Health Index Advisory Group (CHIAG). UK Biobank possesses a Human Tissue Authority (HTA) licence, so a separate HTA licence is not required by researchers who receive samples from the resource, so long as residual samples are destroyed or returned at the end of the research project, and applicants do not transfer the samples to third party premises without the specific approval of UK Biobank. UK Biobank has |

sought generic Research Tissue Bank (RTB) approval, which should cover the large majority of research using the resource. This approach is recommended by the National Research Ethics Service and UK Biobank governing Research Ethics Committee (REC), which approved the application in 2010. Researchers should check the UK Biobank Access Procedures for more detail. FinnGen ethics statement details Patients and control subjects in FinnGen provided informed consent for biobank research, based on the Finnish Biobank Act. Alternatively, separate research cohorts, collected prior the Finnish Biobank Act came into effect (in September 2013) and start of FinnGen (August 2017), were collected based on study-specific consents and later transferred to the Finnish biobanks after approval by Fimea, the National Supervisory Authority for Welfare and Health. Recruitment protocols followed the biobank protocols approved by Fimea. The Coordinating Ethics Committee of the Hospital District of Helsinki and Uusimaa (HUS) approved the FinnGen study protocol Nr HUS/990/2017. The FinnGen study is approved by Finnish Institute for Health and Welfare (permit numbers: THL/2031/6.02.00/2017, THL/1101/5.05.00/2017, THL/341/6.02.00/2018, THL/2222/6.02.00/2018, THL/283/6.02.00/2019, THL/1721/5.05.00/2019, THL/1524/5.05.00/2020, and THL/2364/14.02/2020), Digital and population data service agency (permit numbers: VRK43431/2017-3, VRK/6909/2018-3, VRK/4415/2019-3), the Social Insurance Institution (permit numbers: KELA 58/522/2017, KELA 131/522/2018, KELA 70/522/2019, KELA 98/522/2019, KELA 138/522/2019, KELA 2/522/2020, KELA 16/522/2020 and Statistics Finland (permit numbers: TK-53-1041-17 and TK-53-90-20). The Biobank Access Decisions for FinnGen samples and data utilized in FinnGen Data Freeze 6 include: THL Biobank BB2017_55, BB2017_111, BB2018_19, BB_2018_34, BB_2018_67, BB2018_71, BB2019_7, BB2019_8, BB2019_26, BB2020_1, Finnish Red Cross Blood Service Biobank 7.12.2017, Helsinki Biobank HUS/359/2017, Auria Biobank AB17-5154, Biobank Borealis of Northern Finland_2017_1013, Biobank of Eastern Finland 1186/2018, Finnish Clinical Biobank Tampere MH0004, Central Finland Biobank 1-2017, and Terveystalo Biobank STB 2018001.

Note that full information on the approval of the study protocol must also be provided in the manuscript.

