## [Peer Review File · Nature]

Manuscript Title: Genetic associations of protein-coding variants in human disease

Reviewer Comments & Author Rebuttals

Reviewer Reports on the Initial Version:

Referee #1 (Remarks to the Author):

The authors present a very well-done paper on an important topic. The analyses across hundreds of disease phenotypes and the very large datasets are a real asset. The results will be a valuable resource to the community. I found no technical issues with the way the analysis was done or the way the results were presented. I only have minor comments about this paper:

I see Figure 1c compares the ORs to the MAFs for the novel hits. In general, it would be nice to see a quick summary in the text describing what proportion of the hits were low frequency variants with big ORs, the kind of results that would be helpful to return to an individual, and what proportion were higher frequency variants with very low ORs that were only able to be identified b/c of the very large sample size.

The paper states "We found a statistical enrichment for significant genes in our study to also be approved drug targets (26/482; compared with a background of 569 approved targets/19,955 genes, OR=1.9, $p=0.0024$), which is in line with previous estimates of a higher success rate for drug targets supported by genetics" ... "supporting previous observations that the stronger the genetic association, the higher the likelihood of therapeutic success". This seems circular to me. Doesn't this finding just reflect that previously discovered associations are more likely to have already been tested out in drug trials? If this analysis were done with only the new associations then it might be more interesting.

"As reflected in a recent schizophrenia study⁵⁰, GWAS tend to identify association signals primarily for variants with $MAF > 2\%$, while most variants identified through exome sequencing are ultra-rare ($MAF < 0.01\%$). Of the 975 associations identified in our study, 145 are driven by unique variants in the yet little interrogated rare and low allelic frequency spectrum that is hypothesized to contribute substantially to the "missing heritability" of many human diseases⁵¹." It's not clear to me why the schizophrenia study in particular is cited here, as it's simply a fact that GWAS don't really look at variants with a frequency below 1-2%, and the more rare a variant is, the less power you have to make a discovery with it (given a constant OR). It's also not clear what is meant by "most variants identified through exome sequencing are ultra-rare", as this would entirely depend on the type of analysis performed (common coding variants of course are picked up just fine by exome sequencing, but collapsing analyses of rare variants are usually performed as well). If the goal of this section is to show that variants in the 0.1-2% are overlooked by existing studies, then probably the right way to say that is that GWAS simply don't do MAFs that low, and exome sequencing datasets have so far been more limited in size and lacked power for discovery in this frequency range.

Overall, great paper, look forward to using the results as a resource.

Referee #2 (Remarks to the Author):

This paper describes the results from studying low and rare frequency variants across 744 disease endpoints using exome/imputed genotypes from UK Biobank and the FinnGen study. The focus is on protein coding variation in this really large dataset to determine new associations, report relationships with diseases including mapping of rare variants to GWAS loci to highlight potential

candidate genes, discovery of rare variants causing monogenic disease and their population relevance and relevance as drug targets.

I enjoyed reading this paper and you have done a lot of work with some really nice results covering many disease end points. I thought the study was well designed and have a few comments.

1. For reporting of results it was good to see all the QC checks you did and especially for variants with MAF <1% and as expected there would be more differences between FG and UKB in this allelic range. A review of your results with MAF <1% in ST3 provides a few variants which are quite discrepant In MAF, row 103 for example, CFHR5 the fs variant has a frequency of 0.31% in UKB and 4% in FG – using only a p value of 0.05 and less for support is liberal. Can you comment on your criteria for selecting $P < 0.05$? I also wondered in your general reporting how many associations you found are rare – or low frequency variants your headline reporting does not report results in this way which will be interesting (and a goal of your study?) – how many of these have allelic frequencies that differ significantly across populations. You do describe these results in some detail on page 7 but some further reporting will be interesting.
2. For FG you have used imputed data – and you have been stringent on which variants you included in your analysis. I wondered for many of the positive associations you are reporting (focus here on the rare variants) did you have access to any direct genotypes for some of the variants you report and did you look at the genotype clusters as an extra QC measure? Just as an extra QC measure
3. On page 8 – can you add the MAF of the ODF3 variant?
4. Your manuscript focuses on leveraging coding variant associations for generating new biological insights and you selected AF as one example. In ST3 you report coding variants you have found, and in the check if novel or not only one gene is indicated – ZNF131. Thus, the variants and genes have already been described. I am assuming here a 1 in columns AG and AH refer to a yes novel (there is no legend). The analysis you performed for METTL11B is really interesting and a nice story, at the end of this paragraph you indicate this is the causal gene at this locus – a locus which has many other well established genes with functional data supporting their role in AF (SCN5A). It would be really helpful if you can describe the GWAS locus in more detail – is there only one AF signal at this locus or are there other variants also associated with AF – the GWAS variant maybe tagging variants in other genes at the locus and your work indicates another association and candidate gene?
5. You focused on results from rare variant analyses at the SCN5A and SCN10A locus. This is a well- studied locus with over 30 variants reported with AF/ECG traits and both SCN5A and SCN10A are well established candidate genes. I think the text on this locus could be reduced.
6. HCN4 again another well studied gene and recognised as the candidate at the locus – this text and SCN5A - SCN10A could be a single discussion point?
7. PITX2 – again a well- established candidate gene for AF. It was nice to see the functional work you have done on your new variant, but I thought your results could be placed in better context. The selection of genes you have mostly highlighted for discussion of AF are candidate genes that are recognised and the mechanisms. With your rich dataset of results, were there other loci which you could have highlighted and importantly any new results also from other traits?
8. Over the past week I note there is a new paper on MedRxiv which reports single variant results from UK across 3700 phenotypes (there are some shared co-authors). What are the overlap in findings, can this paper be referred to in some way as there are more exomes and may support some of the results in your paper?
9. Supplementary tables – I do not see any legends for any tables, can these be provided?
10. ST1 – add in the total N. There is no legend, can R5 and R6 be indicated?
11. ST2 – there are some spelling mistakes, for example: Cardiac arrhythmias, please review.
12. ST3 – typos again
13. ST7 – can you also add the RsID to this table and also indicate in column G the source?

14. ST10 – again typographical errors, can you review?
15. Can you provide further details on data sharing – the site and what results?

Referee #3/#4 (Remarks to the Author):

This is the largest association study of protein coding variants identifying known and novel rare and low-frequency protein-coding variants associated with disease. The analysis combines exome sequence data in UK Biobank with imputed genotype data from FinnGen, and may provide a cost-effective approach to increasing power for the detection of rare, disease-associated variants by leveraging MAF enrichment in different populations.

1. The authors state that by combining data from two different populations, whereby rare alleles in one population are enriched in the second population, the authors were able to identify lower frequency variants in regions that have not previously been reported at genome-wide significance. However, there is also gain of power simply due to increased sample size. In Suppl Table 3, only 82 variants had $MAF < 1\%$ in UKB. Most of these variants had larger effect sizes (median odds ratio of 2). If an analysis was done combining the UKB data with another 260,405 European ancestry genotyped individuals imputed to a fully sequenced reference (instead of Finnish samples), would most of these variants still be GWAS significant? In other words, is this just a matter of sufficient power due to increased sample size, or are these variants only identified at GWAS significance because of the higher MAF in the Finnish individuals?

2. For the FinnGen data, how confident are the authors in the accuracy of the imputation for rare variants? The methods do not state if there was any INFO score cut-off applied for filtering of low-quality imputed variants? Could the authors also provide the distribution of INFO scores for $MAF < 1\%$ in FinnGen?

3. The authors derive and validate a theoretical prediction of the improvement in meta-analysis Z-score statistics as a function of sample size, MAF and effect size differences. However, the importance of effect sizes difference is not well reflected in their derivation. In fact, Equation (14) seems to suggest that uplift parameter "alpha" no longer depends on effect sizes, while this is only masked by the parametrisation. We suggest 1) expressing alpha as a function of the odds ratio (OR) in each study and 2) then simplify the expression under the assumption that the OR is constant across studies. This way, the reader will not be confused by statements like "for fixed κ (kappa) IVW uplift α increases with increasing MAF-enrichment", which are incorrect given that kappa is itself a function of MAF-enrichment. We believe that our suggestion should clarify the logic of the argument made there.

4. Pg 7 Line 164 – the authors state that the majority of regions were associated with a single disease cluster. They refer to Ext Data Figure 6. Panel b of this figure shows that most variants are associated with 1-2 biomarker groups. Looking at the biomarker groups, all blood biochemistry measures are classed as one group, despite these biomarkers having very different functions e.g. Vitamin D vs lipids vs inflammatory biomarkers which would be relevant for different disease groups. Lipids would also be more related to anthropometric measures and impedance measures, inflammatory biomarkers related to blood cell counts, urate and creatinine would be more related to blood pressure. Therefore, the number of true pleiotropic associations may be under-estimated. Re-grouping the blood biochemistry biomarkers according to function would make much more sense, rather than a single Blood Biochemistry group. For example see categories listed by UKB https://www.ukbiobank.ac.uk/media/oiudpjqa/bcm023_ukb_biomarker_panel_website_v1-0-aug-2015-edit-2018.pdf.

5. Similarly, Page 11 Line 242, 47 regions were associated with 5 or more biomarker categories including loci such as APOE. But the anthropometry measures, impedance measures and lipids could be considered functionally as one groups, given the causal association between lipids and

anthropometric measures instead of 3 separate groups (vertical pleiotropy as opposed to horizontal pleiotropy). ABCG5 is also listed as a locus associated with 5 or more biomarker categories, but it is only associated with blood biochemistry traits.

6. Supplementary table 10 - rs121913502 (15:90088702:C:T) has a huge effect size (beta=22 in FG) with myeloid leukemia. A lookup of this variant in ClinVar shows it is annotated as pathogenic/likely pathogenic (31 May 2016) for acute myeloid leukemia. But in Suppl Table 3, it shows as NA under ClinVar pathogenicity and 1 as Novel ClinVar variant.

[https://www.ncbi.nlm.nih.gov/clinvar?term=\(\(29755\[AlleleID\]\)OR\(362867\[AlleleID\]\)\)](https://www.ncbi.nlm.nih.gov/clinvar?term=((29755[AlleleID])OR(362867[AlleleID])))

Please re-check ClinVar annotations for all variants to ensure none have been missed and are not labelled as novel when previous evidence is available.

7. There have been some large multi-ancestry exome-sequence studies for specific disease published e.g. T2D (<https://www.nature.com/articles/s41586-019-1231-2>), serum urate (<https://pubmed.ncbi.nlm.nih.gov/30315176/>) and Alzheimer's disease (<https://www.nature.com/articles/s41380-018-0112-7>). Does this analysis replicate these findings, and would these variants have been replicated by UKB alone or only when FinnGen and UKB are combined? Also are variants identified here identified in these studies?

8. Page 8 Line 178 - "152 had previously been linked..." could you add the % in brackets i.e. 28.5%. "For 45 of these, the associated..." could you add % i.e. 29.6%

9. For the variants that were associated with a disease cluster that was different than the phenotype reported in ClinVar, could you provide some examples of how different were the ClinVar phenotypes from the disease cluster? Is there any evidence from mouse studies or drugs side-effects for pleiotropic effects of the genes?

10. In Figure 1C it would be useful to distinguish which of the blue dots were significant in UKB only and those that were significant in FG only.

11. Pg 5 Line 107 - the description of how a distinct region is defined is a little confusing. Could this be written more clearly.

12. Winner's curse can lead to inflated effect sizes that the large effects reported for some of these variants could potentially mislead replication studies, which will be looking for larger effect sizes.

13. Is it possible to compare the proportion of variance explained for a few traits by coding SNPs vs non-coding SNPs since genome-wide data is available for both samples?

14. Page 7 Line 152 - The authors state that 35% of region-disease cluster associations have not been previously reported at $p < 5e-8$. Previous GWAS sample sizes may have been smaller and therefore less powered to detect at genome-wide significance. Is there a way to see if these variants have suggestive association in previous studies ($p < 1e-5$)?

15. It would be useful to know how many distinct genes were associated with disease (outside the HLA region) and how does this compare to the number of distinct regions i.e. how often were coding variants in 2 adjacent genes merged outside the HLA region? Also, of the associated genes, how many had more than one independently associated coding variant?

Author Rebuttal to Initial Comments:

Editor comments

We did not specifically enlist a referee to comment on the wet lab assays and have looked over them in-house. For the revision, please make sure to include the following control experiments:

- a Western blot showing PITX2 expression during the luciferase assay

We now provide a Western blot obtained on the day of the luciferase assay, which shows equal expression of PITX2c/WT and PITX2c/Pro41Ser in **Supplementary Figure Pro41Ser 2 (page 26)**. Extended view of the full membranes and films are also shown. We thank you for this suggestion and the clarity this brings to the manuscript.

- quantification of the qPCR levels of the SLC13A3-reporter mRNA

Thank you for allowing us to clarify this point. For Figure 4b, we conducted a reporter gene assay where the reporter construct contains a promoter fraction of the *SLC13A3* gene containing a presumed PITX2c binding element, but it does not encode for SLC13A3 mRNA. The mRNA produced from this construct encodes solely for luciferase, levels of which are quantified in the assay (relative to beta-galactosidase). To avoid confusion, we are now omitting to refer to SLC13A3 in our main manuscript (**Main text updated in lines 446-449**). We also provide more details on the constructs used to monitor luciferase expression in the section **Transactivation assays** within **Supplementary Methods: Functional characterization of PITX2c Pro41Ser (lines 499-509, page 21-22)**. We apologize for any inconveniences caused in your review of this part of the manuscript and hope that this clarification is satisfactory.

Reviewer comments

Referee #1 (Remarks to the Author):

The authors present a very well-done paper on an important topic. The analyses across hundreds of disease phenotypes and the very large datasets are a real asset. The results will be a valuable resource to the community. I found no technical issues with the way the analysis was done or the way the results were presented. I only have minor comments about this paper:

1. I see Figure 1c compares the ORs to the MAFs for the novel hits. In general, it would be nice to see a quick summary in the text describing what proportion of the hits were low frequency variants with big ORs, the kind of results that would be helpful to return to an individual, and what proportion were higher frequency variants with very low ORs that were only able to be identified b/c of the very large sample size.

We thank the reviewer for this suggestion and have added a few sentences to page 6 (**Main text, lines 127-133**) that feature coding variants with big ORs ($\log[\text{OR}] > 2$) which reside in established disease genes and where return to participants might help improve clinical care. We have modified the main text as follows:

“We found 13 associations (across 11 genes) with log odds ratios > 2 (**Figure 1c**). Of these, 12 associated variants had $\text{MAF} < 1\%$, and only the haemochromatosis variant rs1800562 showed frequency ranges traditionally interrogated in GWAS ($\text{MAF} = 7.9\%$ [UKB], 3.7% [FG]). Several variants with large effect sizes reside in well studied

disease genes such as *BRCA1* (breast cancer), *IDH2* (myeloid leukaemia), *VWF* (von Willebrand disease), or *HFE* (disorders of iron metabolism), proposing that carriers could benefit from clinical monitoring for associated conditions.” (Main text, lines 127-133)

We have further updated Figure 1c and now highlight genes with coding associations of large OR ($\log[\text{OR}]>2$) and very low frequencies ($\text{MAF}<0.1\%$).

In order to further leverage our findings towards “the kind of results that would be helpful to return to an individual” we conducted an additional analysis assessing disease-associations for coding variants within ACMG3.0 genes. The American College of Medical Genetics recommends that variants identified within this set of 73 genes in sequencing studies should be considered for being returned to participants since they are medically actionable (Miller et al., 2021; see References). We have summarized this analysis in the Main text, lines 208-218 as such:

“We also assessed the medical actionability of associated genes as defined in the latest American College of Medical Genetics and Genomics (ACMG) guidelines and found 15 coding variants with significant associations in 11 ACMG genes (Supplementary Table 7). 13 of these associations (1 pathogenic [*BRCA1*], 4 conflicting evidence of pathogenicity, 8 benign/likely benign) had prior ClinVar reports to a matching or putatively related condition, and for several our results proposed extended phenotypes. For example, we found that carriers of the rare missense variant rs370890951 (Ile1131Thr; $\text{MAF}=0.097\%$ [UKB], 0.29% [FG]) in *MYBPC3*, in which mutations cause hypertrophic cardiomyopathy, showed an approximately three-fold increased risk ($p=9.8\times 10^{-13}$) for coagulation defects (Supplementary Table 3, Supplementary Table 7). Together, these findings highlight that population-scale analyses like ours can help refine pathogenicity assignments through contributing quantitative, rather than qualitative information on relative disease risks for variant carriers, or establish an “allelic series” for medically actionable genes.” (Main text, lines 206-218)

Since the definition of a “higher frequency variant with very low OR” can be arbitrary and is confounded by variant enrichment and phenotype ascertainment we have decided to keep the narrative on rare, high-impact variants and hope this is satisfactory for the reviewer.

2. The paper states “We found a statistical enrichment for significant genes in our study to also be approved drug targets (26/482; compared with a background of 569 approved targets/19,955 genes, $\text{OR}=1.9$, $p=0.0024$), which is in line with previous estimates of a higher success rate for drug targets supported by genetics” ... “supporting previous observations that the stronger the genetic association, the higher the likelihood of therapeutic success”. This seems circular to me. Doesn't this finding just reflect that previously discovered associations are more likely to have already been tested out in drug trials? If this analysis were done with only the new associations then it might be more interesting.

We thank the reviewer for this comment. Our finding that genes with coding variant associations have a higher likelihood to be the targets of approved drugs is based on both, known and newly identified association signals. Our approach closely follows that of published studies (e.g. Nelson et al 2015, King et al 2019; see References), with the difference that previous studies primarily considered GWAS variants, including those in non-coding regions, while our results are exclusively based on coding variants. While

largely confirmatory, our analyses nail down the observed enrichments to distinct drug target genes and thus add to insights from earlier studies.

Notably, we limited our analyses to approved drugs to guard against the potential bias that genes with disease associations meanwhile are more likely to be tested in active clinical trials. Since developing a new drug typically takes more than 12 years to approval and insights from population genetics have started to impact pharma R&D only recently, we consider the risk of “reverse associations” as very modest.

Nevertheless, according to the reviewer’s suggestion we have conducted the same analysis only with associations identified in our study. As expected, since the number of approved drug targets with new coding variant associations is still small (n=14), our analysis was underpowered to detect significant enrichment (enrichment odds ratio of 1.3, p=0.34). We have thus decided not to feature this result in the manuscript and hope this is acceptable for the reviewer.

3. "As reflected in a recent schizophrenia study⁵⁰, GWAS tend to identify association signals primarily for variants with MAF>2%, while most variants identified through exome sequencing are ultra-rare (MAF<0.01%). Of the 975 associations identified in our study, 145 are driven by unique variants in the yet little interrogated rare and low allelic frequency spectrum that is hypothesized to contribute substantially to the “missing heritability” of many human diseases⁵¹." It's not clear to me why the schizophrenia study in particular is cited here, as it's simply a fact that GWAS don't really look at variants with a frequency below 1-2%, and the more rare a variant is, the less power you have to make a discovery with it (given a constant OR). It's also not clear what is meant by "most variants identified through exome sequencing are ultra-rare", as this would entirely depend on the type of analysis performed (common coding variants of course are picked up just fine by exome sequencing, but collapsing analyses of rare variants are usually performed as well). If the goal of this section is to show that variants in the 0.1-2% are overlooked by existing studies, then probably the right way to say that is that GWAS simply don't do MAFs that low, and exome sequencing datasets have so far been more limited in size and lacked power for discovery in this frequency range.

In our updated manuscript we have removed the schizophrenia example, which we agree has indeed been a too specific example and rephrased the text according to the reviewer’s suggestion without going into this level of detail. The respective paragraph in the Discussion now reads as such:

“Importantly, our study identifies both, pathogenic variants residing in monogenic disease genes to impact the risk for related complex conditions, as well as new, likely causal sentinel variants within GWAS loci in genes with known and novel biological roles in the respective GWAS trait. With this, our study is one of the first to help bridge the gap between common and rare disease genetics across a broad range of conditions and provides support for the hypothesis that the genetic architecture of many diseases is continuous. Of the 975 associations identified in our study, 145 are driven by unique variants in the yet little interrogated rare and low allelic frequency spectrum between 0.1 and 2% that neither GWAS nor sequencing studies have yet been able to thoroughly interrogate across a range of diseases and that is hypothesized to contribute to the “missing heritability” of many human diseases.”
(Main text, lines 470-479)

Overall, great paper, look forward to using the results as a resource.

We are very happy about the reviewer's constructive comments and enthusiasm for our study.

Referee #2 (Remarks to the Author):

This paper describes the results from studying low and rare frequency variants across 744 disease endpoints using exome/imputed genotypes from UK Biobank and the FinnGen study. The focus is on protein coding variation in this really large dataset to determine new associations, report relationships with diseases including mapping of rare variants to GWAS loci to highlight potential candidate genes, discovery of rare variants causing monogenic disease and their population relevance and relevance as drug targets.

I enjoyed reading this paper and you have done a lot of work with some really nice results covering many disease endpoints. I thought the study was well designed and have a few comments.

We thank the reviewer for appreciating our study and these positive remarks.

1. For reporting of results it was good to see all the QC checks you did and especially for variants with MAF <1% and as expected there would be more differences between FG and UKB in this allelic range. A review of your results with MAF <1% in ST3 provides a few variants which are quite discrepant in MAF, row 103 for example, CFHR5 the fs variant has a frequency of 0.31% in UKB and 4% in FG – using only a p value of 0.05 and less for support is liberal. Can you comment on your criteria for selecting P<0.05? I also wondered in your general reporting how many associations you found are rare – or low frequency variants your headline reporting does not report results in this way which will be interesting (and a goal of your study?) – how many of these have allelic frequencies that differ significantly across populations. You do describe these results in some detail on page 7 but some further reporting will be interesting.

We thank the reviewer for raising these important points.

As we show in Supplementary Figure 1, to be included into our meta-analysis a respective variant had to meet a nominal association significance threshold of $p < 0.05$ for the same trait in both UKB and FG. Our choice of a nominal $p < 0.05$ as entry criterion follows the approach of many previous GWAS (see e.g. our earlier study by Sun et al., Nature 2018; <https://www.nature.com/articles/s41586-018-0175-2#Sec8>). Additional requirements for an association to be called significant in our study was a concordant directionality in both cohorts at inclusion, as well as meeting either the commonly employed GWAS significance threshold of $p < 5 \times 10^{-8}$, or an even more stringent Bonferroni corrected threshold of $p < 2 \times 10^{-9}$ in the meta-analysis.

In order to assess our choice of $p < 0.05$ as entry criterion for our meta-analysis more empirically, we conducted additional computational modelling for our revised manuscript. We first estimated (approximately) the number of expected associations under the null of no association. To preserve the genetic LD and correlations between phenotypes, we permuted disease status (across all diseases simultaneously) between individuals in the UKB cohort. We then applied the identical association analysis approach as for our main analyses. Under the null permuted scenario (permuted 10 times as sensitivity analyses without being too computationally cumbersome), we expected to see on average 39 associations (range: 32-47) compared to 975 associations that we actually observed, giving an approximate empirical false discovery rate (~ 39 expected false positives/all 975 positives observed) of ~ 0.04 (< 0.05) on average, and 0.048 (47/975) in a “worst case” scenario of 10 simulations. In conclusion,

our choice of a $p < 0.05$ threshold for variant inclusion (joint with a meta-analysis cut-off of 5×10^{-8}), is also supported by empirical estimates to appropriately control for false positives. This additional analysis and further justification of a $p < 0.05$ is now detailed in **Supplementary information on choice of significance threshold (Supplementary Information, lines 97-122, page 5)**.

We feel that instead of being liberal the significance criteria in our study have actually been fairly strict. For instance, we used a GWAS-wide significance threshold despite testing a much smaller set of coding variants than typical GWAS, or for Bonferroni correction we assumed that all coding variants tested are independent, which is a stringent assumption ignoring LD. We also consider a concordant beta and $p < 0.05$ as entrance criteria a good trade-off between a high robustness of the association results across both cohorts without ignoring too many true positive signals.

Our choice of $p < 0.05$ is supported by the fact that many expected and well-known associations (such as CTLA4 – Graves’ disease; GBA – Parkinson’s disease; CHEK2 – thyroid cancer, Insulin and diabetes) replicate nominally at around this significance level when conducting analyses in either UKB or FG alone. Such associations would have been missed if we had chosen our entry criteria for meta-analysis to be more stringent.

Conversely, it is evident that at a $p < 0.05$ biologically meaningful true positive signals are being missed, for instance, because there is a strongly significant association in one population, but the variant is absent or extremely rare in the respective other (such as for *CFHR5* that the Reviewer points out). In order to allow readers to follow up on such signals, we have included Supplementary Tables 4 and 5 which list all associations that reach $p < 5 \times 10^{-8}$ in each of the two cohorts individually (318 association signals for UKB and 479 for FG). As we will be releasing all summary results underlying our study, the readers will be able to follow up and replicate our results.

As the reviewer correctly points out, the difference in allelic frequencies between the cohorts is a driver behind some of the most compelling scientific insights from our study. Supplementary Table 3 shows that 145 of the 975 new associations are with variants of a MAF $< 2\%$ in at least one of the two cohorts, of which 50 fall into this frequency range in either cohort. For instance, we feature genes with sentinel variants enriched more than 4-fold in either UKB or FG over the respective other cohort in Table 1. For several of these genes our CWAS reveal an “allelic series” and extend known links with single-gene disorders now to complex diseases of population-level relevance.

We have also incorporated the reviewer’s suggestion “I also wondered in your general reporting how many associations you found are rare – or low frequency variants your headline reporting does not report results in this way which will be interesting (and a goal of your study?) – how many of these have allelic frequencies that differ significantly across populations. You do describe these results in some detail on page 7 but some further reporting will be interesting.”. We have added this additional reporting in the main text on page 7. The respective paragraph now reads:

“In comparison, 52 of the in total 534 (9.7%) of the sentinel variants had MAF $< 1\%$ in either UKB or FG, of which 15 and 23 were enriched by > 2 -fold in UKB and FG respectively (**Supplementary Table 6**).” (**Main text, lines 157-160**)”

2. For FG you have used imputed data – and you have been stringent on which variants you included in your analysis. I wondered for many of the positive associations you are reporting (focus here on the rare

variants) did you have access to any direct genotypes for some of the variants you report and did you look at the genotype clusters as an extra QC measure? Just as an extra QC measure

We thank the reviewer for their interest in our detailed variant QC process. As proposed, we have now added a new Supplementary file (ST3_clusterplots_submission2_ForReviewers.pdf) in which we show cluster plots for 190 exemplary variants from ST3. We prepared graphs for variants genotyped in >46 of 51 of batches of the FinnGen genotyping array, with colours representing successfully genotyped alleles and crosses instances where the respective variant call was missing. None of these cluster plots show quality issues that would be particularly concerning.

We have also added the distribution of the INFO scores for all FG data, the significantly associated sentinel variants, as well as for variants with MAF<1% as a multi-panel figure into the **Supplementary Information (Supplementary Figure Imputation info score distribution of FinnGen data, page 4)**. The INFO scores of the sentinel associated imputed variants were all >0.85. We are thus confident about the imputation quality and QC measures underlying our results (see also our response to Reviewer 3, Comment 2).

Note that for both, genotyping and imputation we performed further QC steps at the batch level as detailed in **Supplementary Methods: FinnGen genetic QC details, pages 2-3**).

3. On page 8 – can you add the MAF of the ODF3 variant?

As proposed, we have now added this information to the main text as such:

“we found an *ODF3* missense variant (rs72878024, MAF=7.5% [UKB], 7.7% [FG]) to be associated with risk of ...” (**Main text, line 186**)

4. Your manuscript focuses on leveraging coding variant associations for generating new biological insights and you selected AF as one example. In ST3 you report coding variants you have found, and in the check if novel or not only one gene is indicated – ZNF131. Thus, the variants and genes have already been described. I am assuming here a 1 in columns AG and AH refer to a yes novel (there is no legend). The analysis you performed for METTL11B is really interesting and a nice story, at the end of this paragraph you indicate this is the causal gene at this locus – a locus which has many other well established genes with functional data supporting their role in AF (SCN5A). It would be really helpful if you can describe the GWAS locus in more detail – is there only one AF signal at this locus or are there other variants also associated with AF – the GWAS variant maybe tagging variants in other genes at the locus and your work indicates another association and candidate gene?

We apologize for any confusion regarding our AF vignettes and the Supplementary Table. We have now added legends to all Supplementary Tables in the Supplementary Information section (**Supplementary Table Legends, pages 32-33**; please see also our response to this Reviewer’s question 9).

Our manual curation was performed only for those variants that were denoted in ST3 as completely novel associations, i.e., they showed a **Novel variant-trait association** both, in GWAS Catalog (GC) and PhenoScanner (PS) = 1 AND **Novel gene-trait implication** both GC and PS = 1 AND **Novel ClinVar** = 1 (meaning variants that showed “1” in all columns AA-AE in ST3), to avoid over-reporting our novel findings and stay conservative, without being too cumbersome manually. Thus novel associations at the variant (and its LD proxies) level in a known gene locus would not be reflected in AG and AH columns,

but would be indicated as “1”s under **Novel variant-trait association** (columns AA and AB). We have clarified this in the newly added Supplementary Table legends and in the **Methods (Main text, lines 656-657)**. Indeed, all but one AF gene locus had already been reported previously at either variant level OR gene level. For AF specifically, we have now performed additional manual curation for novel variant associations (**Novel variant-trait association (both columns AA & AB in ST3 = 1)**) and can confirm these associations and vignettes we feature were not previously reported at the variant level or otherwise explained mechanistically.

In our manuscript we feature four novel rare/low frequency coding associations for AF in *METTL11B*, *SCN5A*, *HCN4*, and *SCN10A* (all MAF<5%). To the best of our knowledge, for none of the featured variants or their LD proxies ($r^2>0.2$) associations with AF had been described previously, although all reside within known AF GWAS loci. The wealth of prior GWAS data was one reason why we selected AF to feature what additional insights beyond GWAS our CWAS results would add.

Regarding *METTL11B*, we are unaware of other AF genes in the proximity of this locus. Notably, *METTL11B* is located on chromosome 1, with the lead variant being 1:170166552:A:G (rs41272485), a different locus than *SCN5A*, which resides on chromosome 3. To avoid any confusion, we have carefully reworded the main text to better reflect that some established AF genes (including *SCN5A*) contain [Ala/Pro/Ser]-Pro-Lys motifs, and are thus potential targets of the METTL11B methylase. The respective sentence now reads as follows:

“The group of proteins containing [Ala/Pro/Ser]-Pro-Lys motifs includes several well-established AF genes⁴⁰ such as potassium channels (*KCNA5*, *KCNE4*, *KCNN3*), sodium channels (*SCN5A*, *SCN10A*), *NPPA*, and *TTN*. Our data support METTL11B as the causal gene in this GWAS locus and a relevance for N-terminal [Ala/Pro/Ser]-Pro-Lys methylation in cardiomyocytes for AF.” (**Main text, lines 381-385**)

The previous GWAS association at the *METTL11B* locus was with a non-coding intergenic variant near *METTL11B* and *LINC01142* that, unlike our study, did not resolve this locus to the likely causal gene. We now better highlight this when introducing the METTL11B paragraph as such:

“The AF GWAS sentinel variant rs72700114 is an intergenic variant located between *METTL11B* and *LINC01142* with no obvious candidate gene.” (**Main text, lines 367-368**)

5. You focused on results from rare variant analyses at the *SCN5A* and *SCN10A* locus. This is a well-studied locus with over 30 variants reported with AF/ECG traits and both *SCN5A* and *SCN10A* are well established candidate genes. I think the text on this locus could be reduced.

6. *HCN4* again another well studied gene and recognised as the candidate at the locus – this text and *SCN5A* - *SCN10A* could be a single discussion point?

There is a connection between the reviewer’s questions 5 & 6 and hence we respond to these together. We agree with the reviewer in that these loci have been well-characterized before and as suggested we have therefore shortened discussion of our findings on these genes in the main text (**Main text, lines 387-414**).

Nevertheless, and as discussed in our response to this Reviewer’s Comment #4, we believe our coding variant associations add additional evidence and insights to these loci. For example, in the recent large-

scale AF GWAS by Nielsen et al, <https://www.nature.com/articles/s41588-018-0171-3>, the authors linked protein altering variants (ST8 in their paper) to the AF loci. This identified the *SCN10A* missense variant rs6795970, which we replicate. In our paper, however, we identified on top of this additional rare coding variants in *SCN5A* at the *SCN5A-SCN10A* GWAS locus (and also in *HCN4* at the *HCN4-REC114* locus) which we on top of AF we link to concordant changes in pulse rate. To improve readability as recommended, we have shortened the main text on this vignette, but we retain a summary of our findings in the **Supplementary Information (Supplementary Discussion on coding associations in *SCN5A-SCN10A* and *HCN4-REC114* AF loci, pages 19-20)**

We now further provide in the Supplement an update on recent findings showing that *SCN10A* plays a key role in the neural modulation of atrial activity and thereby AF inducibility (**Supplementary Discussion on coding associations in *SCN5A-SCN10A* and *HCN4-REC114* AF loci**). On the other hand, the *SCN5A* site identified in our study is associated with cardiac rhythmic disorders appearing with dilated cardiomyopathy. As our PheWAS indicate, this finding could be particularly interesting since the association is with protection from AF, which will need to be elucidated further in future functional studies.

7. PITX2 – again a well-established candidate gene for AF. It was nice to see the functional work you have done on your new variant, but I thought your results could be placed in better context. The selection of genes you have mostly highlighted for discussion of AF are candidate genes that are recognised and the mechanisms. With your rich dataset of results, were there other loci which you could have highlighted and importantly any new results also from other traits?

We thank the reviewer for these comments. There are indeed numerous other examples that we could have highlighted, yet given the vast span of our findings, we had to make trade-offs in terms reporting broadly versus conducting deeper dives. The AF use case highlights at the case of several vignettes how our data can be utilized in a range of different ways to more deeply interrogate individual association signals and inform on the genetics of a distinct complex disease. Among others, we provide for this use case (1) bioinformatic follow-up with orthogonal databases (*METTL11B*); (2) evidence of multiple mechanisms of cardiac related channels in one locus (rare variant associations in both *SCN5A* and *SCN10A* which reside in the same GWAS locus); (3) hints at potential mechanisms at the rare coding-variant level with large effects in *SCN5A* and *HCN4* (discussed also in Supplementary Discussion, pages 19-20); (4) multiple paths to AF risk mechanisms via increased and decreased pulse (through biomarker association scans and clustered MR); and (5) functional follow-up on an AF-associated *PITX2* coding variant. Whilst we acknowledge some of the genes discussed are well established candidate genes, in all cases, our findings yield significant insights at the coding variant level beyond the published literature.

We also provide numerous examples beyond AF on how our data can be used. For instance, we (1) uncover new rare variants in coagulation cascade proteins that provide evidence for different causal mechanisms underlying pulmonary embolism; (2) describe novel coding variants in *SLC34A1* that increase the risk for kidney stones and related biomarkers in a substantial fraction of the European population; or (3) reveal a risk for malignancies not previously associated at the population level with variation in *CHEK2*. Rather than conducting deep-dives for each of our novel findings individually, which would clearly be beyond the scope of a single manuscript, we will be releasing the summary data so that the scientific community can make use of our results in either in similar ways as we have highlighted with the selected vignettes or through alternative approaches.

8. Over the past week I note there is a new paper on MedRxiv which reports single variant results from UK across 3700 phenotypes (there are some shared co-authors). What are the overlap in findings, can this paper be referred to in some way as there are more exomes and may support some of the results in your paper?

The reviewer refers to our recent manuscript entitled “*Systematic single-variant and gene-based association testing of 3,700 phenotypes in 281,850 UK Biobank exomes*” by Karczewski et al. (<https://www.medrxiv.org/content/10.1101/2021.06.19.21259117v1>). Both, that other study and our current manuscript here draw from prioritized access to UKB exomes which we have generated as part of the UKB exome sequencing consortium under UKB application number 26041. Notably, in Karczewski et al. we worked from a substantially smaller UKB exome release (282,000 individuals) than we have utilized in our present study, which comprises coding variants extracted from the exomes of ~392,000 UKB participants and through meta-analysis with FG association results from ~653,000 individuals in total. Consequently, our current study is by far better powered to detect associations, especially in the rare and low-frequency spectrum. We show this by also featuring results analyses obtained from the (expanded) 392k UKB exome dataset alone (without FG) for the diseases studied in our manuscript in Supplementary Table 4. Specifically, association testing within UKB individually yielded 318 associations at $p < 5 \times 10^{-8}$, as opposed to 479 associations in FG individually, and 975 associations in the meta-analysis.

Our current study further differentiates from Karczewski et al. in that beyond simply larger sample sizes we benefit from additional power gains through population specific allelic enrichment. IN our revised manuscript we are now demonstrating this through extensive theoretical and empirical simulations as well as consistency with observed data (please see also our response to Reviewer 3, Comment #1). Additionally, through FG as an independent cohort our study provides replication of results and guards against cohort specific associations. A further difference is that our study showcases numerous vignettes, including deep-dives and an example of functional follow-up with novel discoveries and educates readers how such data can be utilized.

We would like to point out that during the revision process of our manuscript another study on the 300k UKB exome dataset has been published which we have updated the Reference for in **Introduction (Main text, line 63)** (Wang et al., 2021 *Rare variant contribution to human disease in 281,104 UK Biobank exomes* Nature [https:// doi.org/10.1038/s41586-021-03855-y](https://doi.org/10.1038/s41586-021-03855-y)). This study differs from our manuscript in similar ways as the Karczewski et al. study.

We have added the Karczewski et al. MedRxiv paper and also updated the references to other discussed UKB studies in our **Introduction (Main text, line 63)**.

9. Supplementary tables – I do not see any legends for any tables, can these be provided?
10. ST1 – add in the total N. There is no legend, can R5 and R6 be indicated?
11. ST2 – there are some spelling mistakes, for example: Cardiac arrhythmias, please review.
12. ST3 – typos again
13. ST7 – can you also add the RsID to this table and also indicate in column G the source?
14. ST10 – again typographical errors, can you review?

We thank the reviewer for thoroughly reviewing Supplementary Tables and flagging. We have now carefully revised the STs and attempted to remedy all typos and abbreviations. We have made the additions the reviewer requested for ST1 and also for ST7 where we added links to the respective ClinVar entries under “Additional ClinVar Info (Source link)” to provide a full reference for the readers.

We have added ST legends, along with explanation of abbreviations used in the STs in Supplementary Information (**Supplementary Table Legends, pages 32-33**)

15. Can you provide further details on data sharing – the site and what results?

Exome source data utilized in this study have been or will be made publicly available via UK Biobank to qualifying researchers, with release of the data tranche on 456,000 individuals underlying our study anticipated for fall 2021 (for updates, see here: <https://www.ukbiobank.ac.uk/enable-your-research/about-our-data/genetic-data>). FG summary level data is released publicly on a bi-annual basis, with FG release #6 that underlies our study also becoming available in fall 2021. All summary association data underlying our results will be made publicly available for use and download on acceptance of publication. The link will be provided in the published manuscript under “Data availability”.

Referee #3/#4 (Remarks to the Author):

This is the largest association study of protein coding variants identifying known and novel rare and low-frequency protein-coding variants associated with disease. The analysis combines exome sequence data in UK Biobank with imputed genotype data from FinnGen, and may provide a cost-effective approach to increasing power for the detection of rare, disease-associated variants by leveraging MAF enrichment in different populations.

We would like to thank the reviewers for their insightful comments and suggestions which have positively impacted the manuscript.

1. The authors state that by combining data from two different populations, whereby rare alleles in one population are enriched in the second population, the authors were able to identify lower frequency variants in regions that have not previously been reported at genome-wide significance. However, there is also gain of power simply due to increased sample size. In Suppl Table 3, only 82 variants had MAF<1% in UKB. Most of these variants had larger effect sizes (median odds ratio of 2). If an analysis was done combining the UKB data with another 260,405 European ancestry genotyped individuals imputed to a fully sequenced reference (instead of Finnish samples), would most of these variants still be GWAS significant? In other words, is this just a matter of sufficient power due to increased sample size, or are these variants only identified at GWAS significance because of the higher MAF in the Finnish individuals?

The reviewers’ make an excellent point. We have addressed their question via two additional sets of analyses: (1) using real UKB and FG data; and (2) via a realistic simulation study. In doing so:

“we demonstrate both theoretically and in practice that gains in power due to allele enrichment remain even after adjusting for power gains due to increased sample size (**Supplementary Figure MAF enrichment on Z-scores, Extended Data Figure 5d**).”
(Main text, lines 150-152)

Our additional analysis (1) was inspired by the reviewers’ suggestion of “combining the UKB data with another 260,405 European ancestry genotyped individuals imputed to a fully sequenced reference (instead of Finnish samples)”. To do this – within the limitations of the available data – we partitioned the N=392,814 UKB samples into two cohorts: (i) a “base” cohort consisting of $N_1= 132,409$ individuals;

and (ii) an “additional” cohort consisting of the remaining $N_2 = 260,405$ individuals (As suggested, the number of additional samples match the $N = 260,405$ FG individuals in our meta-analysis. We thus refer to this cohort as the FG “sample size matched” cohort).

To assess the relative improvement of meta-analysis Z-scores due to allele enrichment *and* an increased cohort size, against improvements owing to increased cohort size alone, we computed results for two meta-analyses and assessed their ratios. Specifically, we computed $Z_{UKBxFG}/Z_{UKBxUKB}$, where UKBxFG denotes the UKB base cohort combined with FG individuals and UKBxUKB denotes the UKB base cohort combined with the UKB ‘sample size matched’ cohort. The values of $\log_{10}[Z_{UKBxFG}/Z_{UKBxUKB}]$ as a function of allele fold enrichment were concordant with our theoretical expectations that allele enrichment provides additional gains in power over and above increased sample size (**Extended Data Figure 5d**). Note that in this Figure, we log transformed the ratio so >0 reflects a Z-score that is higher in UKBxFG than in UKBxUKB, whereas a Z-score <0 reflects the inverse. Higher fold enrichment (FE) in FG leads to more significant associations in UKBxFG vs UKBxUKB (higher $\log(Z)$ ratio)). Our simulation results thus support systematic power gains due to allele enrichment rather than simply due to sample size (**Extended Data Figure 5d**). A similar trend is also seen for FE in UKB where more negative $\log-Z$ ratio is observed in UKBxUKB vs UKBxFG. The discussion of our approach and results have been added to **Supplementary Information (pages 10-13, pages 16-17)** and as **Extended Data Figure 5d**.

Our additional analysis (2) modifies the simulation protocol detailed in **Supplementary Information** to match the 3-cohort design of analysis (1) above. That is:

“We further investigate the relative gain in IVW Z-scores across a broader range of enrichment values. For this, we modified our simulation strategy detailed in section **Simulations of MAF enrichment effect on inverse-variance weighted meta-analysis Z-scores** to match the cohort analyses above. Specifically, we followed the identical simulation protocol, yet now simulated results from a UKBxFG and separately UKBxUKB meta-analysis, followed by computing the ratio of IVW Z-scores. Results are presented in **Supplementary Figure MAF enrichment on Z-scores**. Again, our theoretical predictions of the relative IVW uplift closely match the simulated results (**Supplementary Figure MAF enrichment on Z-scores (c)**). For instance, when considering a 5-fold MAF enrichment in the FG study, our theoretically predicted estimates (equation (12)) from the simulation study closely approximate the observed estimates (**Supplementary Figure MAF enrichment on Z-scores (d) and Extended Data Figure 5d**). Our simulations further support our findings that studies involving cohorts with MAF enriched designs are likely to provide (potentially significant) additional power gains relative to gains achieved by increasing sample size alone.” (**Supplementary Information, lines 311-324**)

In conclusion, our theoretical, empirical and observed results all suggest power gains from allele population enrichment on top of gains from increased sample size.

2. For the FinnGen data, how confident are the authors in the accuracy of the imputation for rare variants? The methods do not state if there was any INFO score cut-off applied for filtering of low-quality imputed variants? Could the authors also provide the distribution of INFO scores for $MAF < 1\%$ in FinnGen?

As we state in the Introduction section, “the distinct Finnish haplotype structure is characterized by large blocks of co-inherited DNA in linkage disequilibrium and an enrichment for alleles that are rare in other populations, but can still be confidently imputed from genotyping data even in the rare and ultra-rare allele frequency spectrum” (Main text, lines 67-69). We did impose an INFO cut-off (>0.6) as we now clarify in Methods under “Coding variant selection” (Main text, line 585) and “Disease endpoint association analyses (“info”, now capitalized)” (Main text, line 623). Further imputed QC measures are detailed in **Supplementary Information (page 3)**. Additionally, we have added the distribution of FG INFO scores for all FG data, the significantly associated sentinel variants, as well as for all associated variants with MAF<1% as a multi-panel figure into the **Supplementary Information (page 4)**. The INFO scores of the sentinel associated imputed variants were all >0.85. Overall, we are very confident in the imputation quality and QC measures underlying our results (see also our response to Reviewer 2, Comment 2).

3. The authors derive and validate a theoretical prediction of the improvement in meta-analysis Z-score statistics as a function of sample size, MAF and effect size differences. However, the importance of effect sizes difference is not well reflected in their derivation. In fact, Equation (14) seems to suggest that uplift parameter “alpha” no longer depends on effect sizes, while this is only masked by the parametrisation. We suggest 1) expressing alpha as a function of the odds ratio (OR) in each study and 2) then simplify the expression under the assumption that the OR is constant across studies. This way, the reader will not be confused by statements like “for fixed κ (kappa) IVW uplift α increases with increasing MAF-enrichment”, which are incorrect given that kappa is itself a function of MAF-enrichment. We believe that our suggestion should clarify the logic of the argument made there.

We thank the reviewers for their helpful suggestions and apologize for the confusion. As suggested, 1), we have extended the derivation of our estimator to now include (and thus highlight the importance of) effect sizes on the improvement of meta-analysis Z-scores (c.f., **equation (17) in the Supplementary Information**). The reviewers rightly highlight dependencies between variables in the calculation of IVW uplift. We have therefore removed the confusing sentence about fixing the variable κ (kappa).

Further, in the new **equation (16) (c.f. Supplementary Information)** we show that the log odds ratio can be approximated as a function of disease prevalence and MAF, hence the reviewers’ suggestion 2), i.e., assuming that the OR is constant across studies, while helpful, unfortunately doesn’t bypass the issue that, even after assuming the OR is constant across studies, changes in MAF enrichment can still (feasibly) affect κ . We have modified the text to acknowledge this and in doing so clarify our logic as suggested:

“Equation (17) highlights the influence of disease prevalence (π_i), study sample size (N_i), MAF enrichment ($\frac{MAF_2^*}{MAF_1^*}$ and $\frac{MAF_2}{MAF_1}$) and the log odds-ratio (LOR_i), i.e., regression effect size b_i , on IVW uplift α from studies $i \in \{1,2\}$. Owing to some dependencies between variables, e.g., equation (16) relates the log odds ratio to MAF and disease prevalence, it is difficult to make general statements about expected uplift based on increasing MAF enrichment, while attempting to fix the value of other variables. Broadly speaking, however, our theoretical, simulated and observed results indicate that IVW uplift α increases with increasing MAF-enrichment (**Extended Data Figure 5 (a-d), Supplementary Figure 2 and Supplementary Figure MAF enrichment on Z-scores**).” (Supplementary Information, lines 279-287)

4. Pg 7 Line 164 – the authors state that the majority of regions were associated with a single disease cluster. They refer to Ext Data Figure 6. Panel b of this figure shows that most variants are associated with 1-2 biomarker groups. Looking at the biomarker groups, all blood biochemistry measures are classed as one group, despite these biomarkers having very different functions e.g. Vitamin D vs lipids vs inflammatory biomarkers which would be relevant for different disease groups. Lipids would also be more related to anthropometric measures and impedance measures, inflammatory biomarkers related to blood cell counts, urate and creatinine would be more related to blood pressure. Therefore, the number of true pleiotropic associations may be under-estimated. Re-grouping the blood biochemistry biomarkers according to function would make much more sense, rather than a single Blood Biochemistry group. For example see categories listed by UKB

https://www.ukbiobank.ac.uk/media/oiudpiqa/bcm023_ukb_biomarker_panel_website_v1-0-aug-2015-edit-2018.pdf.

5. Similarly, Page 11 Line 242, 47 regions were associated with 5 or more biomarker categories including loci such as APOE. But the anthropometry measures, impedance measures and lipids could be considered functionally as one groups, given the causal association between lipids and anthropometric measures instead of 3 separate groups (vertical pleiotropy as opposed to horizontal pleiotropy). ABCG5 is also listed as a locus associated with 5 or more biomarker categories, but it is only associated with blood biochemistry traits.

Due to the overlap in scope between questions 4 and 5, we reply to both questions simultaneously. We had grouped biomarkers according to the main UKB categories of measurement modality/group and thank the reviewers for providing the link to a more refined biochemistry category list. As per the reviewers' recommendation, we have updated categorizations to now also include more granular UKB biochemistry subgroupings (see revised and updated Supplementary Table 8, Column E and Supplementary Table 9). The main text has been updated as such:

“Additional biochemistry subgroupings were based on UKB biochemistry subcategories:

https://www.ukbiobank.ac.uk/media/oiudpiqa/bcm023_ukb_biomarker_panel_website_v1-0-aug-2015-edit-2018.pdf.” (Main text, lines 670-673)

Accordingly, we have updated Extended Data Figure 6c to better reflect the distributions of associated biomarkers using the newly defined biochemistry subgroupings. We have, however, decided to retain our original less granular categorization for main Figure 3a, primarily to aid visual interpretability of our results. We hope this is acceptable to the reviewers.

In response to comments on specific examples in question 5, we have removed *ABCG5* and *MC1R* from the list of loci discussed in the Main text (**c.f. line 269**) and manually checked the remaining loci for evidence of pleiotropy across the previous and updated groupings. For *APOE*, despite the complexities of grouping biomarkers (as we explain below) and potential vertical pleiotropy around lipid and anthropometric categories, the locus is still associated with liver, renal, CRP, HB1Ac, cancer, bone and joint biochemistry biomarkers, pulse rate, spherical power (eye), and red and white cell indices.

We agree with the reviewers that classifying biomarkers a priori is a complex task that is difficult to perform in both, a systematic automated fashion or through manual curation as there are multiple ways to classify. Groupings can be done by measurement modalities and techniques such as impedance vs physical measures vs lipid biochemistry, or by disease systems. However, there's a large degree of

overlap and correlation using any type of classification as the reviewers highlight, which makes identifying a gold standard approach very challenging. As one example, Alkaline phosphatase (ALP) serves to illustrate the point. ALP is classified herein as bone (and joint) group, following the categories listed by UKB, but is clinically often part of a liver function test panel as ALP can be a sign of liver dysfunction and cholestatic diseases. Additionally, elevated ALP related to bone may also reflect primary and secondary cancers affecting bone, blood cells (lymphomas and leukaemias), kidneys (renal cell carcinoma) – hence, assigning a group for ALP is context specific, with no obvious gold standard. If we were to assign multiple groupings to one biomarker, we run the risk of over-inflating pleiotropic assessments. Consequently, a fine-scale quantification of pleiotropy is difficult and typically not unambiguous. Nevertheless, we are confident that our results are indeed indicative of pleiotropy for the respective loci. As part of our contribution, we will make available the sentinel variant biomarker results also of the non-significant ($p < 1 \times 10^{-6}$) associations, so readers will be able to further customize our data towards their respective definitions and needs.

6. Supplementary table 10 - rs121913502 (15:90088702:C:T) has a huge effect size (beta=22 in FG) with myeloid leukemia. A lookup of this variant in ClinVar shows it is annotated as pathogenic/likely pathogenic (31 May 2016) for acute myeloid leukemia. But in Suppl Table 3, it shows as NA under ClinVar pathogenicity and 1 as Novel ClinVar variant. [https://www.ncbi.nlm.nih.gov/clinvar?term=\(\(29755\[AlleleID\]\)OR\(362867\[AlleleID\]\)\)](https://www.ncbi.nlm.nih.gov/clinvar?term=((29755[AlleleID])OR(362867[AlleleID]))) Please re-check ClinVar annotations for all variants to ensure none have been missed and are not labelled as novel when previous evidence is available.

We thank the reviewers for noticing this mis-annotation. To confirm, we are assuming that the reviewers are referring to Supplementary tables 3 and 7 (i.e., ST3 and ST7) ClinVar annotations, rather than ST10. We have corrected the oversight and can confirm that we have rechecked all other ClinVar phenotype trait mappings. For associations that are novel variant as well as novel gene associations (columns AA-AE = 1 in ST3), we performed additional manual curation to check whether they have been previously reported. In this case, the *IDH2* association with myeloid leukaemia was picked up as being previously reported in our manual curation.

In addition, we are now providing a new column M (“Additional ClinVar Info”) in ST7 which contains the link to the original ClinVar entries for full clarity and for reference.

7. There have been some large multi-ancestry exome-sequence studies for specific disease published e.g. T2D (<https://www.nature.com/articles/s41586-019-1231-2>), serum urate (<https://pubmed.ncbi.nlm.nih.gov/30315176/>) and Alzheimer’s disease (<https://www.nature.com/articles/s41380-018-0112-7>). Does this analysis replicate these findings, and would these variants have been replicated by UKB alone or only when FinnGen and UKB are combined? Also are variants identified here identified in these studies?

We have reviewed the named publications and manually assessed all associations reported as significant with the findings reported in our study. If a reported variant had $p < 0.05$ with the respective trait in either FG or UKB, we performed specific variant association testing and meta-analysis. We summarize our findings in “Additional Tables 1-4”, which we provide for the reviewers and that can be summarized as such:

T2D (Flannick et al):

12 protein coding variants reported in this study as associated with T2D (based on their Extended Data Table 1) were interrogated also in our study. Of these we replicated (at $p < 5 \times 10^{-8}$) 9 associations in the meta-analysis, the same 9 in UKB alone, and 7 in FG alone (with the two remaining variants replicating in FG at $p < 1 \times 10^{-6}$). The remaining three variants, of which two were in the same gene (*SLC16A11*), did not replicate at the chosen significance level. All the betas were in concordant directions between our meta-analysis and Flannick et al. We now make these results available as Additional Table 1.

Notably, we also compared the associations identified in our study against the T2D Knowledge Portal (<https://t2d.hugeamp.org/>) to which Flannick et al. contributed and that contains large collections of meta-analyzed multi-ancestry T2D GWAS. We found matching variants for 32 of 34 sentinel associations reported in our study. Of these, all replicated nominally at $p < 0.05$, and 26 replicated at $p < 5 \times 10^{-8}$ (see Additional Table 2). Again, the betas were concordant and highly correlated between T2D KP and our analysis.

Alzheimer's disease (AD) (Bis et al.):

We assessed the multiple testing-corrected genome-wide significant associations from Bis et al (their Supplementary Table S3). All six reported coding-variant associations replicated in our meta-analysis at $p < 1.3 \times 10^{-4}$ (see Additional Table 3). The *APOE* and *APOC4* variants replicated beyond GWAS significance in both UKB and FG individually, as well as the meta-analysis. The *TREM2* variant rs75932628 had a $p = 0.23$ in FG and $p = 2.1 \times 10^{-4}$ in UKB, with $p = 9.9 \times 10^{-5}$ in the meta-analysis. Its failure to replicate is likely due to the fact that this variant is rare and highly depleted in FG (MAF=0.035%) and still rare in UKB (MAF=0.32%). The remaining three variants (rs28399654, rs28399653, rs3208856) were all at least nominally significant in both FG and UKB, with the meta-analysis boosting the associations. Once again, the betas were concordant.

We did not conduct coding-wide replication analyses for serum urate levels since our primary CWAS are focused on disease endpoints.

Varying sample sizes, allelic enrichment or depletion in different ethnicities, ancestry differences and differences in disease ascertainment between cohorts among others may all feasibly affect replication. Despite this, it is reassuring to see the vast majority of our findings here are highly aligned with findings from large, independent multi-ancestry GWAS meta-analyses. In conclusion, based on the high replication rate for the two selected diseases, we are thus confident that the findings in our CWAS study on 744 disease endpoints are well in line with associations identified through classical single-disease GWAS.

8. Page 8 Line 178 - "152 had previously been linked..." could you add the % in brackets i.e. 28.5%. "For 45 of these, the associated..." could you add % i.e. 29.6%

We have updated and added these suggestions. The respective sentence now reads as such:

"Of the 534 distinct variants with significant disease associations in our study ($p < 5 \times 10^{-8}$), 152 (28.5%) had previously been linked to diseases in ClinVar. For 46 (30.3%) of these variants, the associated disease cluster matched with a previously reported phenotype in ClinVar." (**Main text, lines 192-195**)

Note: 46 instead of 45 updated due to the *IDH2* association being previously associated with multiple myeloma in ClinVar.

9. For the variants that were associated with a disease cluster that was different than the phenotype reported in ClinVar, could you provide some examples of how different were the ClinVar phenotypes from the disease cluster? Is there any evidence from mouse studies or drugs side-effects for pleiotropic effects of the genes?

We have attempted, via manual clinical expert curation, to link putative related associated diseases to ClinVar conditions, with a short brief description of the clinical connection. Results from phenotype matching are now included as an additional column K “Putative related traits for unmatched traits” in ST7.

We have modified the main text to highlight this as such:

“For the remaining 106 ClinVar-listed variants, 29 (27.4%) showed associations to conditions putatively related to those listed in ClinVar (**Supplementary Table 7 and Methods**).” (Main text, lines 198-200)

Determination of related phenotypes requires a degree of clinical judgement which may vary between assessors, but we are confident that our phenotype curation, that has been conducted independently by two experienced medical professionals, should serve well as approximation.

The reviewers make an interesting point about the use of mouse studies and drug side effects for assessing pleiotropy. However, the datasets capturing a broad set of phenotypes concomitantly to do such analyses systematically are currently limited, and it would be challenging to utilize results for generalizable statements. This is certainly an area that should be actively investigated with newer methods in future studies.

10. In Figure 1C it would be useful to distinguish which of the blue dots were significant in UKB only and those that were significant in FG only.

As suggested, we have updated **Figure 1c** to incorporate this change.

11. Pg 5 Line 107 - the description of how a distinct region is defined is a little confusing. Could this be written more clearly.

We appreciated the reviewers’ point and clarified our definition of a distinct region:

“For overlapping genetic regions that are associated with multiple disease endpoints (pleiotropy), to be conservative in reporting the number of associations we merged the overlapping (independent) regions to form a single distinct region (indexed by the Region ID column in **Supplementary Table 3**).” (Main text, line 644-647)

The definition of genetic regions/loci associated with traits become complex in the context of multiple outcomes, especially when some outcomes are related/correlated. Our definition keeps to the conservative side to avoid inflated reports of genetic regions. Specifically, we follow a similar approach to that employed in our previous large-scale pQTL study (Sun et al., Nature 2018; <https://www.nature.com/articles/s41586-018-0175-2>), where analogously to multiple diseases tested

here, there are multiple proteins, and overlapping regions associated with multiple traits in both cases are counted as one region rather than multiple ones. We believe our more conservative reporting, whilst more complex, provides added information and a more objective ‘accounting’ of our discoveries, avoiding inflated reporting of associations.

12. Winner’s curse can lead to inflated effect sizes that the large effects reported for some of these variants could potentially mislead replication studies, which will be looking for larger effect sizes.

We agree with a risk for winner’s curse in any genetic study. However, since the main findings we report are replicated in two independent cohorts where genetic data has been generated on two distinct platforms, and since inclusion into our meta-analysis requires at least nominal significance in each study individually, we feel that our study is considerably more protected from winner’s curse than most large single cohort GWAS.

13. Is it possible to compare the proportion of variance explained for a few traits by coding SNPs vs non-coding SNPs since genome-wide data is available for both samples?

This is a really interesting point that deserves to be thoroughly investigated in follow-up studies that will systematically integrate the coding associations reported here together with non-coding associations across a range of diseases. In practice, we expect it to be challenging to unambiguously separate out signals that are uniquely driven by non-coding relative to coding variants owing to LD, which will likely need to happen on a locus-by-locus basis. Whilst interesting, we feel this is beyond the scope and focus of this study and warrants its own separate investigation in the future. We will be releasing our summary results to allow other researchers to perform additional analyses to help approximate the variance explained.

14. Page 7 Line 152 – The authors state that 35% of region-disease cluster associations have not been previously reported at $p < 5 \times 10^{-8}$. Previous GWAS sample sizes may have been smaller and therefore less powered to detect at genome-wide significance. Is there a way to see if these variants have suggestive association in previous studies ($p < 1 \times 10^{-5}$)?

For our reporting in our cross-reference with previously published disease GWAS, we used $p < 5 \times 10^{-8}$ as cut-off as this is the most widely adopted GWAS significance threshold. As such, many summary statistics in GWAS Catalog and PhenoScanner may only contain genome-wide significant results. Nonetheless, we have additionally performed this analysis cross-referencing GWAS Catalog and PhenoScanner with reviewer suggested $p < 1 \times 10^{-5}$. As expected, we observed a slight decrease, 184/620 (30%) compared to 216/620 (35%), in distinct region-disease cluster associations (at $p < 5 \times 10^{-8}$) that had not previously been reported.

For consistency and interpretability, we keep the reporting in the main text, and we appreciate the reviewers’ good point of many previous studies being smaller and less-powered.

15. It would be useful to know how many distinct genes were associated with disease (outside the HLA region) and how does this compare to the number of distinct regions i.e. how often were coding variants in 2 adjacent genes merged outside the HLA region? Also, of the associated genes, how many had more than one independently associated coding variant?

We thank the reviewers for this suggestion. We have added a short text describing the number of distinct genes associated with disease, as well as a histogram (now featured in **Extended Data Figure 6a**) on how many genes with associations are within each distinct region (per disease cluster; excluding MHC). The vast majority of regions have single gene associations as expected with ~8% where 2 or more adjacent genes were merged (outside the MHC). We have modified the manuscript as follows:

“Mapping associations to genes, we found a total of 482 unique genes associated with the 148 disease clusters. Approximately 92% of the associated regions for each disease cluster (excluding the MHC) harbour a single gene with coding associations (**Extended Data Figure 6a**).” (Main text, lines 176-178)

As there are multiple ways to estimate the number of independent coding associations, we did not perform specific conditional analysis in this study, which goes beyond the main findings and examples our manuscript focuses on. As the summary data will be made available, individual investigators will be able to perform LD-based pruning/summary conditional analysis, e.g., by using GCTA or similar methods on a locus by locus basis. Especially for certain loci which may be complex, individual investigators may wish to choose to model the dependencies between associations through various approaches, some of which may be biologically informed.

Reviewer Reports on the First Revision:

Referee #1 (Remarks to the Author):

The authors have addressed my concerns.

Referee #2 (Remarks to the Author):

Thank you for your detailed responses to all of the points I raised, you have done a lot of additional work with clarifies many of the questions I raised and the other reviewers. In response to my first point, thank you for performing additional analysis to support your threshold of $P < 0.05$. The addition of computational modelling you did and now added to the supplementary material will be informative for other researchers I am sure and it supports your approach. I also appreciate the addition of new tables and providing more information on the QC of rare variants. I had raised a few queries on your discussion of variants in known candidate genes for AF. The revisions you have made to the manuscript address my queries on this and illustrate the motivation and the movement of some of the results to the supplementary material balances out reporting across all the new variant discoveries in the paper.

Referee #3/#4 (Remarks to the Author):

Thank you to the authors for sufficiently addressing all my comments. I have no further comments.

As with other published, large exome sequence studies, the work presented here will be a very useful resource for the research community, and I urge the the authors to consider enabling easy querying of the results e.g. a web interface such as Genebass, in order to enable the international research community to fully leverage this data.

Author Rebuttals to First Revision:

Referee #1 (Remarks to the Author):

The authors have addressed my concerns.

We are pleased that all concerns have been addressed.

Referee #2 (Remarks to the Author):

Thank you for your detailed responses to all of the points I raised, you have done a lot of additional work with clarifies many of the questions I raised and the other reviewers. In response to my first point, thank you for performing additional analysis to support your threshold of $P < 0.05$. The addition of computational modelling you did and now added to the supplementary material will be informative for other researchers I am sure and it supports your approach. I also appreciate the addition of new tables and providing more information on the QC of rare variants. I had raised a few queries on your discussion of variants in known candidate genes for AF. The revisions you have made to the manuscript address my queries on this and illustrate the motivation and the movement of some of the results to the supplementary material balances out reporting across all the new variant discoveries in the paper.

We thank the reviewer's remarks.

Referee #3/#4 (Remarks to the Author):

Thank you to the authors for sufficiently addressing all my comments. I have no further comments.

As with other published, large exome sequence studies, the work presented here will be a very useful resource for the research community, and I urge the the authors to consider enabling easy querying of the results e.g. a web interface such as Genebass, in order to enable the international research community to fully leverage this data.

We thank the reviewers' remarks. We have provided all summary association results; an additional file with just the nominal significant ($p < 0.05$) associations in UKB and FG; and an accompanying README file. These files have been deposited at a public repository and are available for download at: <https://doi.org/10.5281/zenodo.5571000>. This URL has been added to Data Availability section.

We have also been in communications with GWAS Catalog and Open Targets who have both declared their willingness to host our data following formal publication. These portals should meet the recommendations from reviewers 3/4 for a web interface for easy querying of results.